# LFQ: Logit-aware Final-block Quantization for Boosting the Generation Quality of Low-Bit Quantized LLMs

**Jung Hyun Lee** [* 1]  **June Yong Yang** [* 2]  **Jungwook Choi** [3]  **Eunho Yang** [1 4]

## Abstract

As large language models continue to scale, low-bit weight-only post-training quantization (PTQ) offers a practical solution to their memory-efficient deployment. Although block-wise PTQ is capable of matching the full-precision (FP) baseline on basic language modeling and understanding, its quality is degraded for *generative* tasks—especially at longer responses and extended chains of thought, which is critical in boosting task accuracy. We attribute this shortfall to two factors: (i) the omission of the unembedding layer (the LM head) in block-wise optimization and (ii) the reliance on the mean squared error (MSE) objective. Both factors cause the token probability distribution of the quantized model to misalign with that of the FP model, yielding notable accuracy drops on text generation benchmarks. To rectify the discrepancy, we introduce *Logit-aware Final-block Quantization (LFQ)*, a simple yet effective enhancement to block-wise PTQ that quantizes the final Transformer block by minimizing the cross-entropy between the logits of the FP model and those of its quantized counterpart. By aligning token probabilities at the logit level in the final block, LFQ consistently improves the accuracy of complex generation tasks over state-of-the-art block-wise PTQ across diverse model families, while maintaining parity with FP baselines on language modeling and understanding.

[1]Kim Jaechul Graduate School of AI, KAIST, Daejeon, South Korea [2]LG AI Research, Seoul, South Korea [3]Hanyang University, Seoul, South Korea [4]AITRICS, Seoul, South Korea. Correspondence to: Jung Hyun Lee <onliwad101@kaist.ac.kr>, June Yong Yang <laoconeth@gmail.com>.

*Proceedings of the 43rd International Conference on Machine Learning*, Seoul, South Korea. PMLR 306, 2026. Copyright 2026 by the author(s).

## 1. Introduction

The evident success of large language models (LLMs) (Grattafiori et al., 2024; Qwen et al., 2025; Team et al., 2025) is largely attributed to their ever-increasing number of parameters (Kaplan et al., 2020). However, the proportionally increasing memory footprint of the model significantly impedes the cost-effective deployment of LLMs. Not only is a large model difficult to fit in commercial devices, but the serving cost of the model also increases sharply with the model size. To this end, quantization has been widely adopted to increase the inference efficiency of LLMs by employing lower precision data types.

Recently, weight-only quantization (Frantar et al., 2023; Lin et al., 2024) has emerged as a particularly attractive methodology due to its high compression ratio and effective preservation of model quality. By quantizing the LLM weights into low-precision but retaining difficult-to-quantize activations in full precision (FP), memory pressure is effectively relieved while reducing the accuracy degradation. Low-bit weight-only quantization is obtained either via quantization-aware training (QAT) or post-training quantization (PTQ). Although Liu et al. (2025) shows that QAT is capable of restoring the degraded accuracy even under sub-4-bit settings, the computational resource required to conduct QAT makes it prohibitively memory-intensive and time-consuming. On the other hand, layer-wise PTQ can be conducted with a relatively small amount of resources, but suffers from model quality degradation.

Block-wise PTQ (Lee et al., 2023; Shao et al., 2024; Cheng et al., 2024; Lee et al., 2025b; Chen et al., 2025) strikes an effective balance between the two ends, achieving effective and efficient degradation recovery. By minimizing the mean squared error (MSE) between the outputs of an FP Transformer block and those of its quantized counterpart, cross-layer dependencies within each transformer block are accounted for, recovering the performance comparable to FP baselines on tasks such as language modeling (e.g. WikiText2 (Merity et al., 2016)) and general natural language understanding (e.g. MMLU (Hendrycks et al., 2021)).

However, relatively little attention has been shed on the

degradation of *generation* quality caused by the low-bit quantization of LLMs. It is particularly alarming considering the increasing trend towards generating longer responses for increased task performance. Emerging large reasoning models (DeepSeek-AI et al., 2025; Aggarwal & Welleck, 2025) scale inference-time compute to produce extended chains of thought, thereby achieving higher accuracy on complex multi-step reasoning tasks. As this trend toward generating more tokens continues in pursuit of increased accuracy, serving costs rise sharply, necessitating a strong demand for an efficient quantization method that can maintain the generation quality of FP baselines.

In this work, we uncover that the standard block-wise PTQ approach—while effective at language modeling and understanding—suffers from the degradation of generation quality. The limitation is attributed to the fact that block-wise PTQ only preserves the quality of output activations of a transformer block, rather than preserving the fidelity of the next-token sampling *distribution*. Specifically, (i) existing block-wise PTQ methods completely ignore the unembedding layer (also known as the LM head), and (ii) rely on the MSE as optimization objective. Even when the MSE between the outputs of a quantized block and its FP counterpart is minimized, the actual probabilities assigned to plausible tokens can be perturbed, producing substantial shifts in distribution. Such misalignment is less observable on language understanding tasks, which does not involve autoregressive generation, but becomes pronounced in long-form generation as compounding probability distortions steer the generation trajectory away from the FP baseline.

Motivated by the observations, we propose *Logit-aware Final-block Quantization (LFQ)*, which enables low-bit quantized LLMs to achieve performance close to FP baselines on text generation tasks such as IFEval (Zhou et al., 2023), GSM8K (Cobbe et al., 2021), MATH500 (Lightman et al., 2023), and AIME (AIME, 2025). Unlike standard block-wise PTQ, LFQ quantizes the final Transformer block by minimizing the cross-entropy loss between the logits of the FP model and those of its quantized counterpart. Specifically, all Transformer blocks from the first to the penultimate are quantized by minimizing the MSE between the outputs of the FP and quantized blocks, while the final block is optimized using cross-entropy at the logit level, aligning token probabilities with the FP model and thereby reproducing the token prediction probabilities of the FP model. Thanks to its simple design, LFQ can be seamlessly applied to existing block-wise approaches. Moreover, LFQ consistently improves the generation quality of block-wise PTQ methods, while maintaining performance comparable to FP baselines on language modeling and understanding tasks.

Our contribution is threefold:

- To the best of our knowledge, we are the first to

show that the conventional block-wise PTQ objective— minimizing MSE at intermediate outputs—does not align with reproducing the token predictions of FP models, thereby inducing non-negligible accuracy gaps between FP baselines and their low-bit quantized counterparts on text generation tasks.

- We propose Logit-aware Final-block Quantization (LFQ), which quantizes the final Transformer block by minimizing the cross-entropy between the logits of the FP model and its quantized counterpart, consistently improving the generation quality of low-bit quantized LLMs across existing block-wise PTQ methods.

- We validate LFQ across diverse models—including instruction-tuned, reasoning, and Mixture of Experts (MoE) models—on text generation benchmarks such as IFEval, GSM8K, MATH500, and AIME. We further evaluate LFQ on WikiText2 and MMLU to ensure that it performs comparably to, and in some cases better than, existing block-wise PTQ techniques on language modeling and understanding tasks as well.

## 2. Problem Statement

Block-wise PTQ progressively quantizes each Transformer block by minimizing the mean squared error (MSE), from the first to the final block. In this section, we focus our attention on the final block, which is distinctive from the other blocks as it is directly attached to the LM head that produces the token sampling distribution. Below, we provide a brief overview of notations and assumptions used for illustrative purposes throughout this paper.

Let $\boldsymbol{W}_{\text{FP}}, \boldsymbol{W}_q \in \mathbb{R}^{c_{in} \times c_{out}}$ denote the full-precision (FP) final Transformer block and its quantized counterpart, and let $\boldsymbol{X} \in \mathbb{R}^{L \times c_{in}}$ represent the input to the final block, where $L$ is the sequence length. Let $\mathcal{V}$ denote the vocabulary, with size $V = |\mathcal{V}|$. The LM head is then defined as $\boldsymbol{W}_{\text{Head}} \in \mathbb{R}^{c_{out} \times V}$. For illustrative purposes only, however, we restrict the vocabulary to $\mathcal{V} = \{t_1, t_2\}$, so that $\boldsymbol{W}_{\text{Head}} \in \mathbb{R}^{c_{out} \times 2}$. Unless otherwise specified, we omit the normalization layer between the final block and the LM head for simplicity.

First, to illustrate that minimizing the MSE between the outputs of a FP final Transformer block and its quantized counterpart can adversely affect the generation quality of low-bit quantized LLMs, we consider the case where $c_{\text{out}} = 2$. The final block is quantized by minimizing $\|\boldsymbol{X}\boldsymbol{W}_{\text{FP}} - \boldsymbol{X}\boldsymbol{W}_q\|_F^2$, yielding $\boldsymbol{X}\boldsymbol{W}_q = [0.7, 0.3]$ for $\boldsymbol{X}\boldsymbol{W}_{\text{FP}} = [0.8, 0.2]$ as an example. However, it is worth noting the following example: When $\boldsymbol{W}_{\text{Head}} = \begin{bmatrix} 0.5 & 0.3 \\ 0.5 & 1.0 \end{bmatrix}$,

$$\boldsymbol{X}\boldsymbol{W}_{\text{FP}}\boldsymbol{W}_{\text{Head}} = \begin{bmatrix} \mathbf{0.5} \\ 0.44 \end{bmatrix}^T \text{ and } \boldsymbol{X}\boldsymbol{W}_q\boldsymbol{W}_{\text{Head}} = \begin{bmatrix} 0.5 \\ \mathbf{0.51} \end{bmatrix}^T.$$

This result implies that the FP model predicts token $t_1$, while its quantized counterpart instead predicts the opposite token, $t_2$. Consequently, even if the final block is quantized to minimize the MSE between $\boldsymbol{X}\boldsymbol{W}_{\text{FP}}$ and $\boldsymbol{X}\boldsymbol{W}_q$, ignoring the LM head during block-wise quantization can lead the quantized model to produce different token predictions from the FP model.

Even when the LM head is considered, minimizing the MSE between the logits of the FP and quantized models does not guarantee identical token predictions. Suppose we obtain

$$\boldsymbol{X}\boldsymbol{W}_q\boldsymbol{W}_{\text{Head}} = \begin{cases} \text{(i)} \begin{bmatrix} 0.4 \\ \mathbf{0.6} \end{bmatrix}^T \text{ for } \boldsymbol{X}\boldsymbol{W}_{\text{FP}}\boldsymbol{W}_{\text{Head}} = \begin{bmatrix} \mathbf{0.6} \\ 0.4 \end{bmatrix}^T \\ \text{(ii)} \begin{bmatrix} \mathbf{0.6} \\ 0.4 \end{bmatrix}^T \text{ for } \boldsymbol{X}\boldsymbol{W}_{\text{FP}}\boldsymbol{W}_{\text{Head}} = \begin{bmatrix} \mathbf{0.9} \\ 0.1 \end{bmatrix}^T \end{cases}$$
$$(1)$$

Then, the corresponding MSE values are given by

$$\|\boldsymbol{X}\boldsymbol{W}_{\text{FP}}\boldsymbol{W}_{\text{Head}} - \boldsymbol{X}\boldsymbol{W}_q\boldsymbol{W}_{\text{Head}}\|_F^2 \qquad (2)$$
$$= \begin{cases} \text{(i) } (\mathbf{0.6} - 0.4)^2 + (0.4 - \mathbf{0.6})^2 = 0.08 \\ \text{(ii) } (\mathbf{0.9} - \mathbf{0.6})^2 + (0.1 - 0.4)^2 = 0.18 \end{cases}.$$

Although the first case (i) yields the smaller MSE, it leads to the opposite token prediction, whereas the second (ii)—despite having the larger MSE—produces the same token prediction as the FP model. This therefore demonstrates that minimizing MSE at the logit level does not necessarily align with reproducing the FP model's token predictions. Consistently, Figure 1 (a) illustrates that standard block-wise PTQ achieves a lower MSE yet predicts a different top-1 token than the FP model, leading to an incorrect answer.

## 3. Method

As discussed in Section 2, ensuring that low-bit quantized LLMs reproduce the token predictions of their FP counterparts requires explicitly accounting for the LM head and replacing MSE in the optimization objective of block-wise PTQ. To this end, we propose *Logit-aware Final-block Quantization (LFQ)*, which quantizes the final Transformer block by minimizing the cross-entropy loss between the logits of the FP model and those of its quantized counterpart.

### 3.1. Logit-aware Final-block Quantization (LFQ)

Even when the LM head is taken into account, minimizing the MSE does not guarantee that the quantized model will predict the same token as the FP model. Since minimizing cross-entropy is equivalent to minimizing KL divergence, and KL divergence is equal to zero if and only if two distributions are identical, minimizing cross-entropy at the logit

level directly encourages the quantized model's token-level distribution to match its FP counterpart. Furthermore, as Bruch (2021) demonstrates, cross-entropy can be used for learning to rank, which would also help the quantized model recover the FP model's top-k token ordering. Accordingly, when optimizing the quantized final Transformer block $\boldsymbol{W}_q$, we minimize the cross-entropy between the FP model's logits and those of the quantized model to align the block-wise PTQ objective with the FP model's token generation. Specifically, while the first through penultimate Transformer blocks are quantized by minimizing the MSE between the outputs of the FP and quantized blocks, the final block is quantized using the following optimization objective:

$$\min_{\boldsymbol{W}_q} \mathcal{L}_{\text{CE}}(\boldsymbol{X}\boldsymbol{W}_{\text{FP}}\boldsymbol{W}_{\text{Head}}, \boldsymbol{X}\boldsymbol{W}_q\boldsymbol{W}_{\text{Head}}) \qquad (3)$$

$$= -\frac{1}{L}\sum_{i,j=1}^{L,V} \sigma(\boldsymbol{X}\boldsymbol{W}_{\text{FP}}\boldsymbol{W}_{\text{Head}})_{i,j} \log(\sigma(\boldsymbol{X}\boldsymbol{W}_q\boldsymbol{W}_{\text{Head}}))_{i,j},$$

where $\sigma(\boldsymbol{Z}) := \frac{\exp(Z_{i,j})}{\sum_{k=1}^{V}\exp(Z_{i,k})}$ for $\boldsymbol{Z} = [Z_{i,j}]_{i=1,j=1}^{L,V}$. We refer to Eq. 3 as "LFQ."

The specific quantization parameters contained in $\boldsymbol{W}_q$ depend on the chosen block-wise PTQ method. For example, when instantiating (a) FlexRound (Lee et al., 2023), (b) OmniQuant (Shao et al., 2024), or (c) Block-AP (Chen et al., 2025), Eq. 3 specializes accordingly as:

(a) $\boldsymbol{W}_q = \boldsymbol{s}_1 \left\lfloor \dfrac{\boldsymbol{W}_{\text{FP}}}{\boldsymbol{s}_1 \odot \boldsymbol{S}_2 \odot \boldsymbol{s}_3} \right\rceil$ \qquad (4)

where $\boldsymbol{s}_1, \boldsymbol{s}_3 \in \mathbb{R}_{>0}^{c_{out} \times \frac{c_{in}}{g}}$, and $\boldsymbol{S}_2 \in \mathbb{R}_{>0}^{c_{out} \times c_{in}}$,
$\Rightarrow$ Eq. 3: $\min_{\boldsymbol{s}_1, \boldsymbol{S}_2, \boldsymbol{s}_3} \mathcal{L}_{\text{CE}}(\boldsymbol{X}\boldsymbol{W}_{\text{FP}}\boldsymbol{W}_{\text{Head}}, \boldsymbol{X}\boldsymbol{W}_q\boldsymbol{W}_{\text{Head}})$.

(b) $\boldsymbol{W}_q = \boldsymbol{h}\left\lfloor \dfrac{\boldsymbol{W}_{\text{FP}}}{\boldsymbol{h}} \right\rceil$ where $\boldsymbol{\gamma}, \boldsymbol{\beta} \in \mathbb{R}_{[0,1]}^{c_{out} \times \frac{c_{in}}{g}}$, \qquad (5)

and $\boldsymbol{h} = \dfrac{\boldsymbol{\gamma}\max(\boldsymbol{W}_{\text{FP}}) - \boldsymbol{\beta}\min(\boldsymbol{W}_{\text{FP}})}{2^b - 1}$
$\Rightarrow$ Eq. 3: $\min_{\boldsymbol{\gamma}, \boldsymbol{\beta}} \mathcal{L}_{\text{CE}}(\boldsymbol{X}\boldsymbol{W}_{\text{FP}}\boldsymbol{W}_{\text{Head}}, \boldsymbol{X}\boldsymbol{W}_q\boldsymbol{W}_{\text{Head}})$.

(c) $\boldsymbol{W}_q = \boldsymbol{s}\left\lfloor \dfrac{\boldsymbol{W}_{\text{FP}}}{\boldsymbol{s}} \right\rceil$ where $\boldsymbol{s} \in \mathbb{R}_{>0}^{c_{out} \times \frac{c_{in}}{g}}$, \qquad (6)

$\Rightarrow$ Eq. 3: $\min_{\boldsymbol{s}, \boldsymbol{W}_{\text{FP}}} \mathcal{L}_{\text{CE}}(\boldsymbol{X}\boldsymbol{W}_{\text{FP}}\boldsymbol{W}_{\text{Head}}, \boldsymbol{X}\boldsymbol{W}_q\boldsymbol{W}_{\text{Head}})$.

Here, $b$ is the bit-width and $g$ is the group size ($g = c_{in}$ for per-channel quantization, and $g = 128$ for group-wise quantization). We refer to Eq. 4, 5, and 6 as "FlexRound+LFQ", "OmniQuant+LFQ", and "Block-AP+LFQ", respectively.

Three points are worth highlighting here. First, since LFQ integrates the LM Head and cross-entropy into the loss objective of standard block-wise PTQ, it is agnostic to the underlying block-wise method and thus can be applied seamlessly. Second, as LFQ optimizes only the final Transformer

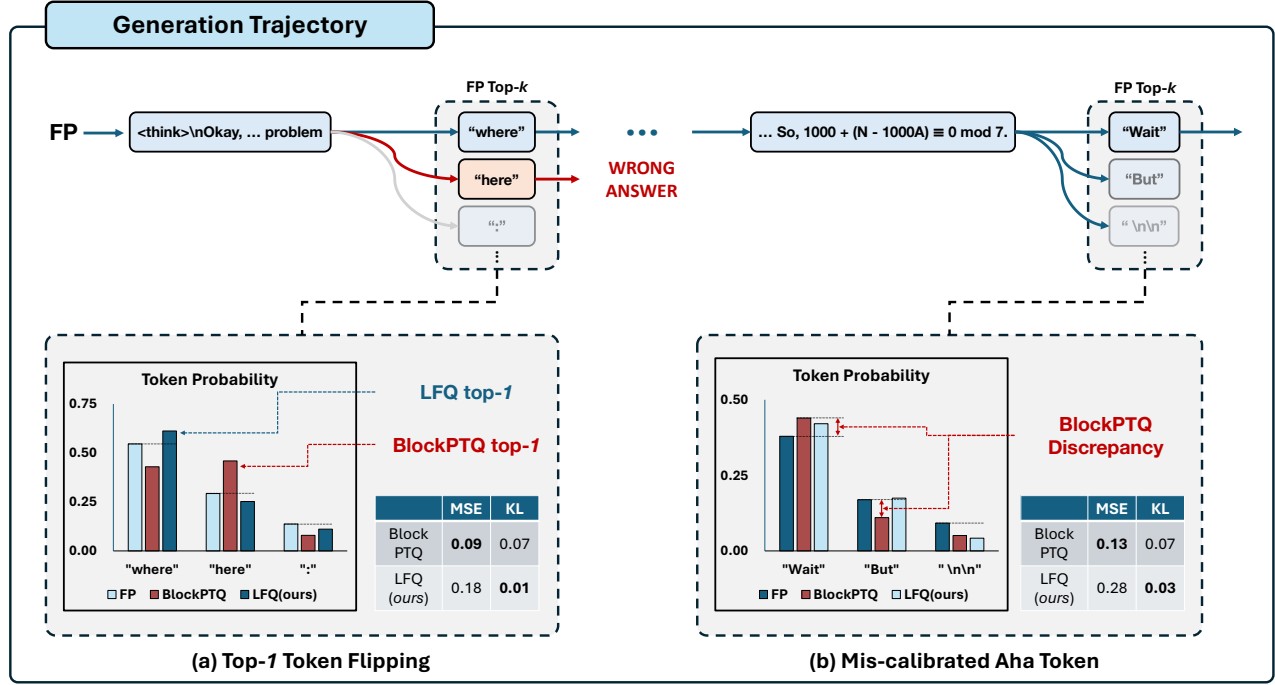

*Figure 1.* Reasoning trajectory of L1-Qwen-7B-Max under greedy decoding on AIME 2024 Problem 28. We compare token-level probability distributions for the FP baseline, block-wise PTQ ("blockPTQ" in this figure), and LFQ (ours) at two instants: (a) the first step where block-wise PTQ's top-1 token diverges from the FP baseline, and (b) the first "aha" moment guiding the reasoning onto the correct path. In (a), block-wise PTQ's top-1 (`"here"`) corresponds to the FP baseline's top-2, yielding an incorrect answer, whereas LFQ's top-1 (`"where"`) matches the FP baseline's top-1 and thus reaches the correct answer. In (b), block-wise PTQ is overconfident in `"Wait"` and underconfident in `"But"`, while LFQ assigns probabilities to these "aha" tokens that remain closer to the FP baseline.

block by minimizing cross-entropy at the logit level, it is memory-efficient and thus can be run on a single GPU like other block-wise PTQ techniques. Third, because LFQ modifies only the optimization objective and leaves the quantization scheme unchanged, LFQ-quantized LLMs remain fully compatible with existing packing/unpacking kernels (e.g., Frantar et al. (2023); Lin et al. (2024); Park et al. (2024)) and can therefore be accelerated without additional effort.

### 3.2. Effect of LFQ on Token Generation

To illustrate that cross-entropy better reproduces the FP model's token predictions than MSE, we revisit the example 1 in Section 2. The corresponding cross-entropy values are given as follows:

$$-\frac{1}{L} \sum_{i,j=1}^{L,V} \sigma(\mathbf{X}\mathbf{W}_{\text{FP}}\mathbf{W}_{\text{Head}})_{i,j} \log(\sigma(\mathbf{X}\mathbf{W}_q\mathbf{W}_{\text{Head}}))_{i,j}$$

$$= \begin{cases} \text{(i)} & -\mathbf{0.6}\log(0.4) - 0.4\log(\mathbf{0.6}) \approx 0.75 \\ \text{(ii)} & -\mathbf{0.9}\log(0.6) - 0.1\log(0.4) \approx 0.55 \end{cases}$$

In contrast to MSE, which is lower in case (i) than in case (ii) despite case (i) predicting the opposite token and case (ii) predicting the same token as the FP model, cross-entropy assigns a smaller value to case (ii) than to case (i). This observation highlights that minimizing the cross-entropy loss at the logit level is essential for guiding low-bit quantized LLMs to align with the FP model's token predictions.

To make this trend concrete, Figure 1 presents a reasoning trajectory generated by L1-Qwen-7B-Max for Problem 28 of AIME 2024 and compares token-level probability distributions for the FP baseline, block-wise PTQ, and LFQ at two key points: (a) the first instance where block-wise PTQ's top-1 token diverges from the FP baseline, and (b) the first "aha" moment that steers the reasoning onto the correct path. In Figure 1 (a), although block-wise PTQ attains a smaller MSE than LFQ, thanks to minimizing cross-entropy at the logit level, LFQ yields a smaller KL divergence from the FP distribution. Consequently, LFQ reproduces the FP model's top-1 token prediction (i.e., `"where"`) and follows the correct trajectory to the right answer, whereas block-wise PTQ diverges and thus fails to solve the problem.

*Table 1.* Performance of Qwen2.5-7B-Instruct and Qwen2.5-14B-Instruct with LFQ under block-wise PTQ (FlexRound, OmniQuant, and Block-AP). Within each PTQ method, the best accuracy is shown in **bold**. "W4" and "W3g128" denote 4-bit per-channel weight-only quantization and 3-bit group-wise quantization (group size 128), respectively. LFQ yields consistent gains in generation quality across block-wise PTQ, while preserving the language modeling and understanding performance of existing methods.

| Method | # Bits | Language Modeling/Understanding | | Text Generation | |
| | | WikiText2 ($\downarrow$) | MMLU ($\uparrow$) | IFEval ($\uparrow$) (greedy) | MATH500 ($\uparrow$) (greedy) |
| --- | --- | --- | --- | --- | --- |
| Qwen2.5-7B-Instruct | BF16 | 6.85 | 73.49 | 70.79 | 74.2 |
| FlexRound | W4 | 7.23 | **72.50** | 69.50 | 72.6 |
| FlexRound+LFQ (Ours) | W4 | **7.21** | 72.48 | **71.35** | **73.4** |
| FlexRound | W3g128 | 7.63 | 70.13 | 66.54 | 65.6 |
| FlexRound+LFQ (Ours) | W3g128 | **7.58** | **70.26** | **67.84** | **68.0** |
| OmniQuant | W4 | 7.73 | **71.00** | 68.21 | 69.8 |
| OmniQuant+LFQ (Ours) | W4 | **7.53** | 70.99 | **69.50** | **71.6** |
| OmniQuant | W3g128 | 8.08 | **68.43** | 68.21 | 63.6 |
| OmniQuant+LFQ (Ours) | W3g128 | **7.91** | 68.39 | **68.58** | **64.4** |
| Block-AP | W4 | 7.87 | 69.60 | 66.73 | 68.0 |
| Block-AP+LFQ (Ours) | W4 | **7.77** | **69.94** | **68.02** | **69.0** |
| Block-AP | W3g128 | 8.70 | **67.09** | 61.00 | 60.0 |
| Block-AP+LFQ (Ours) | W3g128 | **8.18** | 67.06 | **63.77** | **61.8** |
| Qwen2.5-14B-Instruct | BF16 | 5.24 | 78.82 | 79.85 | 78.4 |
| FlexRound | W4 | 5.67 | **77.33** | 77.82 | 76.4 |
| FlexRound+LFQ (Ours) | W4 | **5.62** | 77.31 | **78.00** | **77.2** |
| FlexRound | W3g128 | 6.15 | 75.84 | 75.05 | 69.6 |
| FlexRound+LFQ (Ours) | W3g128 | **6.11** | **75.85** | **77.08** | **71.6** |
| OmniQuant | W4 | 5.93 | 76.64 | 73.94 | 73.4 |
| OmniQuant+LFQ (Ours) | W4 | **5.89** | **76.66** | **75.23** | **75.2** |
| OmniQuant | W3g128 | 6.43 | 75.62 | 74.31 | **70.4** |
| OmniQuant+LFQ (Ours) | W3g128 | **6.36** | **75.73** | **75.42** | 69.8 |
| Block-AP | W4 | 6.23 | 76.84 | 70.79 | 71.6 |
| Block-AP+LFQ (Ours) | W4 | **6.17** | **76.86** | **72.27** | **72.4** |
| Block-AP | W3g128 | 6.81 | **74.58** | 71.72 | 67.0 |
| Block-AP+LFQ (Ours) | W3g128 | **6.69** | **74.58** | **72.46** | **68.0** |

Moreover, because the occurrence of an "aha" moment is pivotal for re-evaluating and correcting an ongoing reasoning trajectory, the extent to which low-bit quantized models track the FP baseline on such "aha" tokens—e.g., `"Wait"` and `"But"`—is a key determinant of their accuracy on complex reasoning benchmarks. As shown in Figure 1 (b), even when the top-1 token for all three models is `"Wait"`, it is noteworthy that block-wise PTQ is overconfident in `"Wait"`, leaving it underconfident in another "aha" token like `"But"`. By contrast, LFQ allocates these probabilities closer to the FP baseline, resulting in not only a smaller KL divergence but also higher accuracy than block-wise PTQ as reported in Table 2.

## 4. Experiment

In this section, we first assess LFQ on Qwen2.5-7B-Instruct and Qwen2.5-14B-Instruct (Qwen et al., 2025) using IFEval and MATH500. We then evaluate LFQ on reasoning models—L1-Qwen-7B-Max (Aggarwal & Welleck, 2025) and DeepSeek-R1-Distill-Llama-8B (DeepSeek-AI et al., 2025)—on MATH500 and AIME 2024 (AIME′24). We also validate LFQ on Qwen3-30B-A3B-Instruct-2507 (Team, 2025) using IFEval and AIME 2025 (AIME′25). Finally, we empirically (i) demonstrate the importance of incorporating the LM head and utilizing cross-entropy in the objective, (ii) validate that quantizing only the final Transformer block via logit-level cross-entropy is sufficient (i.e., it is not a must to quantize multiple final blocks with cross-entropy), and (iii) the comparison of LFQ against LoRA-based quantization error compensation (LQEC). These findings are mainly established on Llama 3.1 8B Instruct (Grattafiori et al., 2024) using IFEval and GSM8K.

We select calibration samples from the beginning of C4 training set (Raffel et al., 2020) in sequential order. We do so to emphasize that LFQ can preserve performance comparable to FP baselines on language modeling (e.g., WikiText-2) and understanding (e.g., MMLU), while consis-

*Table 2.* Performance of L1-Max-Qwen-7B and DeepSeek-R1-Distill-Llama-8B with LFQ under block-wise PTQ (FlexRound). Within the PTQ method, the best accuracy is shown in **bold**. "W4" and "W3g128" denote 4-bit per-channel weight-only quantization and 3-bit group-wise quantization (group size 128), respectively. LFQ yields consistent gains in generation quality, while preserving the language modeling and understanding performance of existing methods. To estimate avg@8 and pass@8, we use temperature 0.6 and top-p 0.95 with a maximum generation length of 4096 tokens. For pass@8, we sample 16 responses per question.

| Method | # Bits | Language Modeling/Understanding | | Text Generation | | |
| | | WikiText2 ($\downarrow$) | MMLU ($\uparrow$) | MATH500 ($\uparrow$) (avg@8) | AIME′24 ($\uparrow$) (greedy) | AIME′24 ($\uparrow$) (pass@8) |
|---|---|---|---|---|---|---|
| L1-Qwen-7B-Max | BF16 | 29.57 | 54.58 | 89.05$\pm$0.74 | 46.67 | 55.30 |
| FlexRound | W4 | 31.20 | **53.43** | 87.45$\pm$0.75 | 30.00 | 51.71 |
| FlexRound+LFQ (Ours) | W4 | **30.44** | 53.10 | **88.40$\pm$0.86** | **43.33** | **55.09** |
| FlexRound | W3g128 | 31.45 | 52.24 | 85.35$\pm$0.64 | 23.33 | 41.85 |
| FlexRound+LFQ (Ours) | W3g128 | **29.46** | **52.53** | **86.50$\pm$0.54** | **30.00** | **45.18** |
| DeepSeek-R1-Distill-Llama-8B | BF16 | 11.85 | 55.69 | 72.53$\pm$1.16 | 30.00 | 30.49 |
| FlexRound | W4 | 12.61 | **54.57** | 70.10$\pm$1.37 | 16.67 | 27.71 |
| FlexRound+LFQ (Ours) | W4 | **12.46** | 54.21 | **71.95$\pm$1.20** | **26.67** | **30.07** |
| FlexRound | W3g128 | 13.80 | 53.61 | 67.00$\pm$0.97 | 10.00 | 15.98 |
| FlexRound+LFQ (Ours) | W3g128 | **13.24** | **54.03** | **69.25$\pm$0.85** | **13.33** | **16.97** |

*Table 3.* Performance of Qwen3-30B-A3B-Instruct-2507 with LFQ under block-wise PTQ (FlexRound). Within the PTQ method, the best accuracy is shown in **bold**. "W4g128" denote 4-bit group-wise quantization (group size 128). LFQ yields consistent gains in generation quality, while preserving the language modeling and understanding performance of existing methods. To estimate pass@1, we use temperature 0.7, top-p 0.8, and top-k 20, and sample 16 responses per question with a maximum generation length of 32768 tokens.

| Method | # Bits | Language Modeling/Understanding | | Text Generation | | |
| | | WikiText2 ($\downarrow$) | MMLU ($\uparrow$) | IFEval ($\uparrow$) (greedy) | AIME′25 ($\uparrow$) (greedy) | AIME′25 ($\uparrow$) (pass@1) |
|---|---|---|---|---|---|---|
| Qwen3-30B-A3B-Instruct-2507 | BF16 | 7.00 | 81.12 | 83.18 | 66.67 | 62.50 |
| GPTQ | W4g128 | 7.32 | 80.15 | 82.26 | 50.00 | 55.63 |
| FlexRound | W4g128 | 7.33 | 80.24 | 82.44 | 53.33 | 57.29 |
| FlexRound+LFQ (Ours) | W4g128 | **7.27** | **80.25** | **82.99** | **60.00** | **58.75** |

tently improving the generation quality of block-wise PTQ. We report perplexity on WikiText2 using a sequence length of 4096, five-shot accuracy on MMLU, prompt-level strict-accuracy on IFEval (following Qwen et al. (2025)), 8-shot accuracy on GSM8K, and zero-shot accuracy on MATH500 and AIME. Further setup details are given in Appendix E.

## 4.1. Qwen2.5-Instruct on IFEval and MATH500

To verify whether LFQ can improve low-bit instruction-tuned LLMs on both natural language instruction following and challenging math word problems, we evaluate LFQ for Qwen2.5-7B-Instruct and Qwen2.5-14B-Instruct on IFEval and MATH500 using greedy decoding. Table 1 shows that, across different quantization configurations, LFQ consistently improves the generation quality of instruction-tuned models quantized by FlexRound, OmniQuant, and Block-AP on both IFEval and MATH500. Consequently, FlexRound+LFQ narrows the gap between 4-bit per-channel models and their FP baselines to within 1 percentage point

(pp) for Qwen2.5-7B-Instruct and within 2 pp for Qwen2.5-14B-Instruct across all benchmarks considered.

## 4.2. Reasoning Models on MATH500 and AIME′24

To test whether LFQ can also perform well for reasoning models that produce long chains of thought by scaling test-time compute, we apply LFQ to L1-Qwen-7B-Max and DeepSeek-R1-Distill-Llama-8B on MATH500 and AIME′24. Given that Table 1 identifies FlexRound+LFQ as the most effective, we focus on FlexRound+LFQ hereafter, unless otherwise noted. Table 2 shows that block-wise PTQ suffers substantial degradation under greedy decoding on AIME′24. In contrast, LFQ nearly matches the FP baseline on AIME′24, indicating that it restores alignment with the FP model's top-1 token predictions. Furthermore, LFQ raises pass@8 to within 0.5 percentage points of the FP baselines. Taken together, these results suggest that LFQ effectively aligns the token-level probabilities of low-bit quantized LLMs with those of their FP counterparts.

*Table 4.* Performance of Llama 3.1 8B Instruct when block-wise PTQ methods (FlexRound, OmniQuant, and Block-AP) are incrementally augmented by (i) incorporating the LM head and (ii) using a logit-level cross-entropy objective in order to quantize the final Transformer block. Within each block-wise PTQ method, the best accuracy is shown in **bold** and the second-best is underlined. Here, all results use 4-bit per-channel weight-only quantization. LFQ (with both LM Head and cross-entropy, ours) yields consistent gains in generation quality across block-wise PTQ, while preserving the language modeling and understanding performance of existing methods.

| Method | LM-Head | Cross-Entropy | Language Modeling/Understanding | | Text Generation | |
| --- | --- | --- | --- | --- | --- | --- |
| | | | WikiText2 ($\downarrow$) | MMLU ($\uparrow$) | IFEval ($\uparrow$) (greedy) | GSM8K ($\uparrow$) (greedy) |
| Llama 3.1 8B Instruct | N/A | N/A | 6.75 | 68.34 | 74.49 | 84.99 |
| FlexRound | X | X | 7.06 | 66.19 | 70.24 | 81.35 |
| FlexRound+LFQ | O | X | 7.08 | 66.75 | 71.53 | 81.58 |
| | O | O | **7.06** | **66.97** | **72.09** | **81.80** |
| OmniQuant | X | X | 7.49 | 64.87 | 70.61 | 78.17 |
| OmniQuant+LFQ | O | X | 7.48 | 64.77 | 71.35 | 78.32 |
| | O | O | **7.47** | **65.48** | **71.35** | **79.76** |
| Block-AP | X | X | 7.76 | 63.24 | 68.58 | 73.84 |
| Block-AP+LFQ | O | X | 7.74 | 63.54 | 68.39 | 74.00 |
| | O | O | **7.69** | **63.77** | **68.76** | **74.45** |

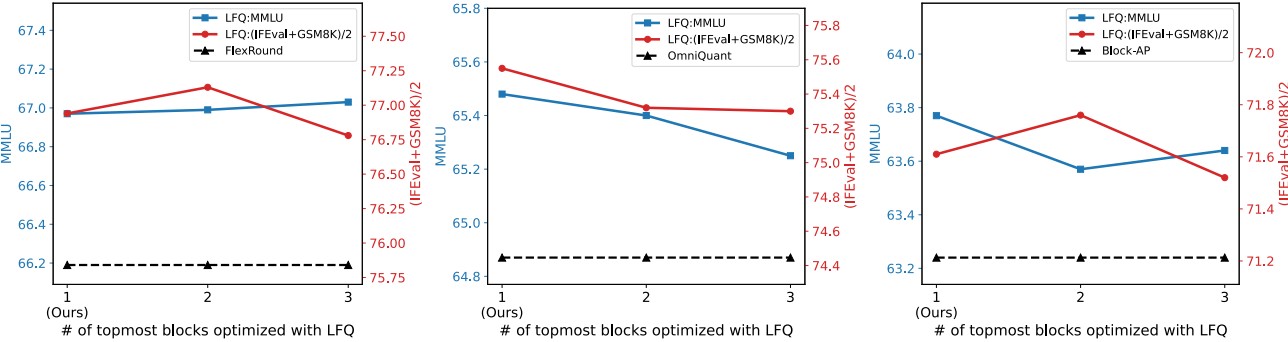

*Figure 2.* Performance of Llama 3.1 8B Instruct as the number of topmost Transformer blocks optimized with LFQ increases from 1 (ours) to 3, with the remaining blocks quantized via standard MSE reconstruction; shown for FlexRound (left), OmniQuant (center), and Block-AP (right). In each subfigure, the left y-axis shows MMLU accuracy, while the right y-axis reports the IFEval+GSM8K average. All results use 4-bit per-channel weight-only quantization. The average of IFEval and GSM8K (expressed as "(IFEval+GSM8K)/2") stays roughly unchanged, regardless of the number of topmost Transformer blocks optimized with LFQ.

### 4.3. Qwen3-30B-A3B-Instruct-2507 on IFEval and AIME′25

We conduct additional LFQ experiments on Qwen3-30B-A3B-Instruct-2507 to demonstrate that LFQ is effective for both MoE and traditional dense architectures. As shown in Table 3, LFQ improves the pass@1 score on AIME′25 by approximately 1.5 pp and the IFEval score by about 0.5 pp, thereby narrowing the performance gap between the quantized MoE model and its FP counterpart.

### 4.4. Ablation Study

**Importance of LM Head and cross-entropy.** To assess the impact of incorporating the LM head and using cross-entropy in the loss objective when quantizing the final block, we incrementally augment existing block-wise PTQ methods (FlexRound, OmniQuant, and Block-AP) with these

components. As shown in Table 4, adding the LM head alone generally improves accuracy on text generation benchmarks (IFEval and GSM8K) as well as on language modeling and understanding. With the LM head in place, employing cross-entropy rather than mean squared error (MSE) yields further gains on text generation tasks. We therefore conclude that leveraging both the LM head and cross-entropy, as in Eq. 3, is essential for boosting the generation quality of low-bit quantized LLMs. We conduct additional ablation study for Qwen2.5-7B-Instruct in Appendix B.

**Sufficiency of quantizing solely the final block via logit-level cross-entropy.** We ask whether applying the logit-level cross-entropy objective to only the final block is sufficient. To test this, we vary the number of topmost Transformer blocks optimized with LFQ (denoted as $k$) while keeping the remaining blocks quantized via standard MSE reconstruction. For example, when $k = 2$, we apply LFQ se-

*Table 5.* Comparison of LFQ (ours) against RILQ, a state-of-the-art LoRA-based quantization error compensation method, on Llama 3.1 8B Instruct using block-wise PTQ (FlexRound, OmniQuant, and Block-AP). Within each PTQ method, the best accuracy is shown in **bold** and the second best is underlined. "W4" denotes 4-bit per-channel weight-only quantization.

| Method | # Bits | Language modeling/understanding | | Text generation | |
| | | WikiText2 ($\downarrow$) | MMLU ($\uparrow$) | IFEval ($\uparrow$) (greedy) | GSM8K ($\uparrow$) (greedy) |
|---|---|---|---|---|---|
| Llama 3.1 8B Instruct | BF16 | 6.75 | 68.34 | 74.49 | 84.99 |
| FlexRound | W4 | 7.06 | 66.19 | 70.24 | 81.35 |
| FlexRound+RILQ | W4 | **6.95** | 66.86 | 71.90 | 80.52 |
| FlexRound+LFQ | W4 | 7.06 | **66.97** | **72.09** | **81.80** |
| OmniQuant | W4 | 7.49 | 64.87 | 70.61 | 78.17 |
| OmniQuant+RILQ | W4 | **7.24** | **66.07** | 71.35 | 78.85 |
| OmniQuant+LFQ | W4 | 7.47 | 65.48 | **71.35** | **79.76** |
| Block-AP | W4 | 7.76 | 63.24 | 68.58 | 73.84 |
| Block-AP+RILQ | W4 | **7.43** | **64.62** | 68.58 | 73.92 |
| Block-AP+LFQ | W4 | 7.69 | 63.77 | **68.76** | **74.45** |

quentially to the penultimate and final blocks. As shown in Figure 2, the average score of IFEval and GSM8K remains almost constant even as $k$ increases; $k = 2$ occasionally yields a marginal gain on that average but at the cost of lower MMLU accuracy. These results indicate that applying LFQ to the final block alone is sufficient and offers the best overall trade-off. We also emphasize in Appendix C that *to which* block LFQ is applied is far more important than *how many* blocks are optimized with LFQ.

**Comparison of LFQ against LQEC.** As LQEC has emerged as a promising approach for mitigating memory bottleneck while recovering task accuracy, we compare LFQ with RILQ (Lee et al., 2025a), a state-of-the-art LQEC method. For a fair comparison, we use the C4 training set as calibration data to initialize LoRA adapters on low-bit quantized models produced by FlexRound, OmniQuant, and Block-AP. In Table 5, RILQ often outperforms LFQ on language modeling (WikiText2) and understanding (MMLU) due to its use of LoRA adapters, but LFQ consistently surpasses RILQ on text generation (IFEval and GSM8K) across all settings. We hypothesize that this stems from the fact that LQEC methods—including RILQ—optimizes MSE, an objective misaligned with matching the FP model's token-level distribution (as elucidated in Section 2). To leverage the strengths of both methods, we also explore their joint application in Appendix D.

## 5. Related Work

It is well known that quantization-aware training (QAT) can match full-precision (FP) accuracy even under sub-4-bit quantization configurations, so it has been applied across domains—from computer vision models (Esser et al., 2020; Lee et al., 2021) to natural language models (Liu et al., 2023;

2025). However, Liu et al. (2025) shows that QAT requires fine-tuning LLMs on billions of tokens at least, which is prohibitively memory-intensive and time-consuming. Consequently, research attention has continued to focus more on advancing post-training quantization (PTQ).

PTQ is commonly divided into layer-wise and block-wise methods. Layer-wise PTQ (e.g., Frantar et al. (2023); Lin et al. (2024)) can be run fast on a single GPU and typically incurs marginal performance degradation on relatively easy downstream tasks (e.g., commonsense reasoning). However, as these techniques do not involve gradient-based optimization, unless task-specific calibration data is utilized, they can suffer substantial accuracy degradation on more challenging benchmarks, particularly text generation (Li et al., 2025). On the other hand, block-wise PTQ approaches (Lee et al., 2023; Shao et al., 2024; Cheng et al., 2024; Lee et al., 2025b; Chen et al., 2025; Park et al., 2025) not only account for cross-layer dependencies within a block but also optimize quantization parameters via gradient-based iterative updates, and therefore often outperform layer-wise PTQ. Notwithstanding, we find that existing block-wise PTQ can still exhibit non-negligible degradation in generation quality.

## 6. Conclusion

We show that block-wise PTQ can degrade generation quality due to (i) omitting the LM head from block-wise optimization and (ii) relying on the MSE objective. To address this, we introduce Logit-aware Final-block Quantization (LFQ), which quantizes the final Transformer block by aligning the quantized model's logits to the FP model's via the cross-entropy loss. Across diverse model families and generation tasks, LFQ consistently improves generation quality over existing block-wise PTQ techniques, while preserving performance on language modeling and understanding.

## Acknowledgements

This work was supported by Institute for Information & communications Technology Planning & Evaluation(IITP)grant funded by the Korea government(MSIT) (RS-2019-II190075, Artificial Intelligence Graduate School Program(KAIST)) and National Research Foundation of Korea (NRF) grant (No.RS-2023-00209060, A Study on Optimization and Network Interpretation Method for Large-Scale Machine Learning) funded by the Korea government (MSIT). This work was supported by the Institute of Information & Communications Technology Planning & Evaluation (IITP) grant funded by the Korea government (MSIT) (No.RS-2026-25507427, Development of Efficient Architectures and Training Techniques for High-Performance Lightweight AI Models).

## Impact Statement

This paper presents work aimed at improving the inference efficiency of large language models. We anticipate several potential societal implications, but we do not believe any require specific emphasis here.

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

## A. Additional Avg@8 Results on IFEval and MATH500 for Qwen2.5-7B-Instruct

*Table 6.* Performance of Qwen2.5-7B-Instruct with LFQ under block-wise PTQ (FlexRound, OmniQuant, and Block-AP). Within each PTQ method, the best accuracy is shown in **bold**. "W4" and "W3g128" denote 4-bit per-channel weight-only quantization and 3-bit group-wise quantization (group size 128), respectively. To estimate avg@8, we use temperature 0.7, top-p 0.8, and top-k 20.

| Method | # Bits | Language Modeling/Understanding | | Text Generation | |
| | | WikiText2 ($\downarrow$) | MMLU ($\uparrow$) | IFEval ($\uparrow$) (avg@8) | MATH500 ($\uparrow$) (avg@8) |
|---|---|---|---|---|---|
| Qwen2.5-7B-Instruct | BF16 | 6.85 | 73.49 | 70.82±0.88 | 73.65±1.09 |
| FlexRound | W4 | 7.23 | **72.50** | 69.64±0.65 | 69.33±1.21 |
| FlexRound+LFQ (Ours) | W4 | **7.21** | 72.48 | **71.44**±0.55 | **70.35**±1.16 |
| FlexRound | W3g128 | 7.63 | 70.13 | 68.30±1.07 | 63.83±0.92 |
| FlexRound+LFQ (Ours) | W3g128 | **7.58** | **70.26** | **69.50**±0.56 | **65.35**±0.76 |
| OmniQuant | W4 | 7.73 | **71.00** | 68.46±0.72 | 68.28±1.00 |
| OmniQuant+LFQ (Ours) | W4 | **7.53** | 70.99 | **69.78**±0.91 | **69.53**±1.00 |
| OmniQuant | W3g128 | 8.08 | **68.43** | 68.23±1.04 | 61.75±1.12 |
| OmniQuant+LFQ (Ours) | W3g128 | **7.91** | 68.39 | **68.63**±0.45 | **63.58**±0.89 |
| Block-AP | W4 | 7.87 | 69.60 | 66.87±0.85 | 65.98±1.42 |
| Block-AP+LFQ (Ours) | W4 | **7.77** | **69.94** | **68.12**±0.97 | **67.25**±1.15 |
| Block-AP | W3g128 | 8.70 | **67.09** | 61.35±1.05 | 58.08±0.82 |
| Block-AP+LFQ (Ours) | W3g128 | **8.18** | 67.06 | **64.03**±0.79 | **61.53**±1.25 |

Table 6 shows that LFQ also improves Avg@8 scores on IFEval and MATH500 for Qwen2.5-7B-Instruct across different block-wise PTQ methods (FlexRound, OmniQuant, and Block-AP) and quantization schemes (W4 and W3g128), demonstrating that LFQ is effective under both greedy and stochastic decoding.

# B. Additional Ablation Studies of LFQ

*Table 7.* Performance of Qwen2.5-7B-Instruct when block-wise PTQ methods (FlexRound, OmniQuant, and Block-AP) are incrementally augmented by (i) incorporating the LM head and (ii) using a logit-level cross-entropy objective in order to quantize the final Transformer block. Within each block-wise PTQ method, the best accuracy is shown in **bold** and the second-best is underlined. Here, all results use 4-bit per-channel weight-only quantization. LFQ (with both LM Head and cross-entropy, ours) yields consistent gains in generation quality across block-wise PTQ, while preserving the language modeling and understanding performance of existing methods.

| Method | LM-Head | Cross-Entropy | Language Modeling/Understanding | | Text Generation | |
| --- | --- | --- | --- | --- | --- | --- |
| | | | WikiText2 ($\downarrow$) | MMLU ($\uparrow$) | IFEval ($\uparrow$) (greedy) | MATH500 ($\uparrow$) (greedy) |
| Qwen2.5-7B-Instruct | N/A | N/A | 6.85 | 73.49 | 70.79 | 74.2 |
| FlexRound | X | X | 7.23 | **72.50** | 69.50 | 72.6 |
| FlexRound+LFQ | O | X | 7.26 | 72.48 | 71.35 | 71.4 |
| | O | O | **7.21** | 72.48 | **71.35** | **73.4** |
| OmniQuant | X | X | 7.73 | 71.00 | 68.21 | 69.8 |
| OmniQuant+LFQ | O | X | **7.29** | **71.02** | 68.95 | 70.6 |
| | O | O | 7.53 | 70.99 | **69.50** | **71.6** |
| Block-AP | X | X | 7.87 | 69.60 | 66.73 | 68.0 |
| Block-AP+LFQ | O | X | 7.92 | 69.75 | 67.28 | 68.4 |
| | O | O | **7.77** | **69.94** | **68.02** | **69.0** |

*Table 8.* Performance of Llama 3.2 3B Instruct when block-wise PTQ methods (FlexRound, OmniQuant, and Block-AP) are incrementally augmented by (i) incorporating the LM head and (ii) using a logit-level cross-entropy objective in order to quantize the final Transformer block. Within each block-wise PTQ method, the best accuracy is shown in **bold** and the second-best is underlined. Here, all results use 4-bit per-channel weight-only quantization. LFQ (with both LM Head and cross-entropy, ours) yields consistent gains in generation quality across block-wise PTQ, while preserving the language modeling and understanding performance of existing methods.

| Method | LM-Head | Cross-Entropy | Language Modeling/Understanding | | Text Generation | |
| --- | --- | --- | --- | --- | --- | --- |
| | | | WikiText2 ($\downarrow$) | MMLU ($\uparrow$) | IFEval ($\uparrow$) (greedy) | GSM8K ($\uparrow$) (greedy) |
| Llama 3.2 3B Instruct | N/A | N/A | 10.14 | 61.34 | 71.72 | 77.48 |
| FlexRound | X | X | 10.72 | **59.93** | 65.80 | 72.40 |
| FlexRound+LFQ | O | X | 10.73 | 59.66 | 65.99 | **73.09** |
| | O | O | **10.72** | 59.84 | **67.28** | 73.01 |
| OmniQuant | X | X | 11.23 | 57.78 | 64.33 | 71.42 |
| OmniQuant+LFQ | O | X | 11.17 | 58.80 | 64.33 | **71.65** |
| | O | O | **11.16** | **58.94** | **66.54** | 71.49 |
| Block-AP | X | X | 11.61 | **56.47** | 62.29 | 66.11 |
| Block-AP+LFQ | O | X | 11.63 | 56.44 | 62.66 | 66.34 |
| | O | O | **11.61** | 56.38 | **63.77** | **66.41** |

## C. Importance of Applying LFQ to the Last Block

To emphasize that *to which* block LFQ is applied is far more important than *how many* blocks are optimized with LFQ, we conduct the following experiments for Llama 3.1 8B Instruct. When $k = 2$, we apply LFQ to the second-to-last block, while for the last block we only minimize $\|XW_{\text{FP}} - XW_q\|_F^2$ without using either the LM head or cross-entropy (denoted as "$k = 2$ except the last block"). When $k = 3$, we apply LFQ sequentially to the third- and second-to-last blocks, and again, for the last block only, we minimize $\|XW_{\text{FP}} - XW_q\|_F^2$ without employing the LM head or cross-entropy (denoted as "$k = 3$ except the last block"). We then compare these settings with the original $k = 2$ and $k = 3$ configurations in Figure 2.

*Table 9.* Comparison of $k = 2$ and $k = 3$ except the last block with the original $k = 2$ and $k = 3$ configurations in Figure 2. "LFQ@Last" indicates whether LFQ is applied to the last block. Similarly, "LFQ@Last-1" and "LFQ@Last-2" indicate whether LFQ is applied to the second-to-last and third-to-last blocks, respectively.

| Method | LFQ@Last | LFQ@Last-1 | LFQ@Last-2 | MMLU (↑) | (IFEval+GSM8K)/2 (↑) |
|---|---|---|---|---|---|
| Llama 3.1 8B Instruct | N/A | N/A | N/A | 68.34 | 79.74 |
| FlexRound | X | X | X | 66.19 | 75.80 |
| + $k = 2$ except the last block | X | O | X | 66.97 | 76.17 |
| + $k = 2$ (Figure 2) | **O** | O | X | **66.99** | **77.13** |
| + $k = 3$ except the last block | X | O | O | 66.98 | 76.15 |
| + $k = 3$ (Figure 2) | **O** | O | O | **67.03** | **76.78** |
| OmniQuant | X | X | X | 64.87 | 74.39 |
| + $k = 2$ except the last block | X | O | X | 65.29 | 74.47 |
| + $k = 2$ (Figure 2) | **O** | O | X | **65.40** | **75.32** |
| + $k = 3$ except the last block | X | O | O | **65.29** | 74.65 |
| + $k = 3$ (Figure 2) | **O** | O | O | 65.25 | **75.30** |
| Block-AP | X | X | X | 63.24 | 71.21 |
| + $k = 2$ except the last block | X | O | X | **63.78** | 71.30 |
| + $k = 2$ (Figure 2) | **O** | O | X | 63.57 | **71.76** |
| + $k = 3$ except the last block | X | O | O | **63.81** | 71.24 |
| + $k = 3$ (Figure 2) | **O** | O | O | 63.64 | **71.52** |

As shown in Table 9, the MMLU score remains nearly unchanged regardless of whether LFQ is applied to the final block. In contrast, when LFQ is not applied to the final block, the average of IFEval and GSM8K (i.e., "(IFEval+GSM8K)/2") consistently drops, approaching the performance level of each underlying PTQ technique. These results indicate that, for improving the generation quality of low-bit quantized LLMs, it is far more critical to apply LFQ to the final block than to simply increase the number of blocks optimized with LFQ.

# D. Combination of LFQ with RILQ

*Table 10.* Comparison of LFQ (ours) against RILQ, a state-of-the-art LoRA-based quantization error compensation method, on Llama 3.1 8B Instruct using block-wise PTQ (FlexRound, OmniQuant, and Block-AP). Within each PTQ method, the best accuracy is shown in **bold** and the second best is underlined. "W4" denotes 4-bit per-channel weight-only quantization.

| | | Language modeling/understanding | | Text generation | |
|---|---|---|---|---|---|
| Method | # Bits | WikiText2 ($\downarrow$) | MMLU ($\uparrow$) | IFEval ($\uparrow$) (greedy) | GSM8K ($\uparrow$) (greedy) |
| Llama 3.1 8B Instruct | BF16 | 6.75 | 68.34 | 74.49 | 84.99 |
| FlexRound | W4 | 7.06 | 66.19 | 70.24 | 81.35 |
| FlexRound+RILQ | W4 | **6.95** | 66.86 | 71.90 | 80.52 |
| FlexRound+LFQ | W4 | 7.06 | **66.97** | 72.09 | **81.80** |
| FlexRound+LFQ+RILQ | W4 | 6.98 | 66.96 | **72.46** | 81.43 |
| OmniQuant | W4 | 7.49 | 64.87 | 70.61 | 78.17 |
| OmniQuant+RILQ | W4 | 7.24 | **66.07** | 71.35 | 78.85 |
| OmniQuant+LFQ | W4 | 7.47 | 65.48 | 71.35 | **79.76** |
| OmniQuant+LFQ+RILQ | W4 | **7.23** | 65.82 | 71.35 | 79.45 |
| Block-AP | W4 | 7.76 | 63.24 | 68.58 | 73.84 |
| Block-AP+RILQ | W4 | 7.43 | **64.62** | 68.58 | 73.92 |
| Block-AP+LFQ | W4 | 7.69 | 63.77 | **68.76** | **74.45** |
| Block-AP+LFQ+RILQ | W4 | **7.43** | 64.53 | 68.58 | 74.22 |

LFQ underperforms RILQ on WikiText2 perplexity (language modeling) and MMLU accuracy (language understanding), while outperforming RILQ on IFEval and GSM8K (text generation), as shown in Table 5. However, we emphasize that LFQ (quantization objective) and RILQ (LoRA addition) address orthogonal aspects of the problem rather than competing with each other. Because LFQ can be readily combined with RILQ in a complementary manner, we therefore explore their joint application to leverage the strengths of both methods.

LFQ + RILQ performs comparably to RILQ on WikiText2 perplexity and MMLU accuracy, while achieving results close to LFQ on IFEval and GSM8K. This indicates that LFQ + RILQ can effectively inherit the strengths of both techniques.

# E. Experimental Setting of LFQ

We sweep the LFQ learning rate as follows: $\{5e-4, 1e-3\}$ with FlexRound; $\{1.5e-3, 2e-3, 5e-3\}$ with OmniQuant; and $\{1e-5, 2e-5, 3e-5\}$ with Block-AP. Across all block-wise PTQ methods, calibration samples are drawn from the beginning of C4 training set (Raffel et al., 2020) in sequential order: $800$ for Llama-3.2-3B-Instruct; $600$ for Qwen2.5-7B-Instruct and L1-Max-Qwen-7B; $550$ for Llama-3.1-8B-Instruct; $512$ for DeepSeek-R1-Distill-Llama-8B and Qwen3-30B-A3B-Instruct-2507; and $400$ for Qwen2.5-14B-Instruct. We also sample calibration sequences from the beginning of C4 training set in sequential order, using a sequence length of 4096 tokens for Qwen3-30B-A3B-Instruct-2507 and 2048 tokens for the other models. For the remaining hyperparameters, we follow the experimental settings recommended in prior work (Lee et al., 2023; Shao et al., 2024; Chen et al., 2025).

# F. Learning Rate Sensitivity of LFQ

*Table 11.* Performance of Qwen2.5-7B-Instruct with LFQ under block-wise PTQ (FlexRound, OmniQuant, and Block-AP). Within each block-wise PTQ method, the best accuracy is shown in **bold** and the second-best is underlined. "W4" indicates 4-bit per-channel weight-only quantization. "lr" denotes the learning rate used in LFQ.

| Method | # Bits | Language Modeling/Understanding | | Text Generation | |
| | | WikiText2 ($\downarrow$) | MMLU ($\uparrow$) | IFEval ($\uparrow$) (greedy) | MATH500 ($\uparrow$) (greedy) |
| --- | --- | --- | --- | --- | --- |
| Qwen2.5-7B-Instruct | BF16 | 6.85 | 73.49 | 70.79 | 74.2 |
| FlexRound | W4 | 7.23 | **72.50** | 69.50 | 72.6 |
| + LFQ (lr = $5e-4$) | W4 | **7.20** | 72.41 | 71.16 | 73.2 |
| + LFQ (lr = $1e-3$) | W4 | 7.21 | 72.48 | **71.35** | **73.4** |
| OmniQuant | W4 | 7.73 | **71.00** | 68.21 | 69.8 |
| + LFQ (lr = $1.5e-3$) | W4 | 7.57 | 70.93 | **69.87** | 70.6 |
| + LFQ (lr = $2e-3$) | W4 | **7.53** | 70.99 | 69.50 | **71.6** |
| Block-AP | W4 | 7.87 | 69.60 | 66.73 | 68.0 |
| + LFQ (lr = $2e-5$) | W4 | 7.77 | **69.94** | 68.02 | **69.0** |
| + LFQ (lr = $3e-5$) | W4 | **7.75** | 69.85 | **68.76** | 68.4 |

As demonstrated in Table 11, LFQ consistently improves the generation quality of low-bit quantized LLMs regardless of the choice of learning rate.

# G. Comparison of WikiText2 and C4 as calibration datasets for LFQ

*Table 12.* Performance of Qwen2.5-7B-Instruct with LFQ under block-wise PTQ (FlexRound, OmniQuant, and Block-AP). Within each block-wise PTQ method, the best accuracy is shown in **bold** and the second-best is underlined. "W4" indicates 4-bit per-channel weight-only quantization. The calibration dataset used in LFQ is either WikiText2 or C4.

| Method | # Bits | Language Modeling/Understanding | | Text Generation | |
| | | WikiText2 ($\downarrow$) | MMLU ($\uparrow$) | IFEval ($\uparrow$) (greedy) | MATH500 ($\uparrow$) (greedy) |
|---|---|---|---|---|---|
| Qwen2.5-7B-Instruct | BF16 | 6.85 | 73.49 | 70.79 | 74.2 |
| FlexRound | W4 | 7.23 | **72.50** | 69.50 | 72.6 |
| + LFQ (WikiText2) | W4 | **7.15** | 72.37 | 70.98 | 73.0 |
| + LFQ (C4) | W4 | 7.21 | 72.48 | **71.35** | **73.4** |
| OmniQuant | W4 | 7.73 | **71.00** | 68.21 | 69.8 |
| + LFQ (WikiText2) | W4 | **7.47** | 70.92 | **69.87** | 71.6 |
| + LFQ (C4) | W4 | 7.53 | 70.99 | 69.50 | **71.6** |
| Block-AP | W4 | 7.87 | 69.60 | 66.73 | 68.0 |
| + LFQ (WikiText2) | W4 | **7.73** | 69.85 | **68.39** | 68.8 |
| + LFQ (C4) | W4 | 7.77 | **69.94** | 68.02 | **69.0** |

Table 12 shows that LFQ enhances the generation quality of low-bit quantized LLMs irrespective of the choice of calibration dataset.

