# OpenReview forum: "LFQ: Logit-aware Final-block Quantization for Boosting the Generation Quality of Low-Bit Quantized LLMs"
_ICML.cc/2026/Conference — ICML 2026 regular_

### Official Review · Reviewer_iVTV · 2026-02-28

**Soundness:** 3
**Presentation:** 3
**Significance:** 2
**Originality:** 2
**Overall Recommendation:** 4
**Confidence:** 3

**Summary:**

The paper identifies two critical factors contributing to the generation quality degradation in standard block-wise post-training quantization (PTQ): (i) the omission of the unembedding layer (LM head) during optimization and (ii) the reliance on the mean squared error (MSE) objective. To address this, the authors propose Logit-aware Final-block Quantization (LFQ), a method that quantizes the final Transformer block by minimizing the cross-entropy (CE) loss between the logits of the full-precision (FP) model and its quantized counterpart. The proposed method is  compatible with existing PTQ frameworks and its effectiveness is evaluated across various models and  tasks.

**Compliance With Llm Reviewing Policy:**

Affirmed.

**Final Justification:**

My final recommendation is 4 (Weak Accept).

The paper addresses a practically important issue in post-training quantization and proposes a simple, technically sound improvement that can be integrated into existing block-wise PTQ methods. I find the main idea reasonable and the empirical results generally supportive, especially given the consistency across different backbones and quantization settings.

The rebuttal was helpful and partially addressed my concerns by adding stochastic decoding results and qualitative examples. However, I still think the evaluation is somewhat limited for fully supporting the claim that CE-based optimization better preserves the full output distribution, particularly in realistic long-form generation settings.  So I maintain my Weak Accept recommendation.

**Key Questions For Authors:**

please see the weaknesses.

**Limitations:**

yes

**Strengths And Weaknesses:**

### Strengths：
* The paper identifies a limitation in current block-wise PTQ methods, highlighting that minimizing MSE  does not necessarily preserve the output token distribution of the FP baseline.
* The proposed LFQ can be easily integrated into existing block-wise PTQ methods by optimizing the CE loss on the logits of the final Transformer block.

### Weaknesses:

* A major claim of the paper is that CE loss aligns the entire token distribution better than MSE. However, almost all primary results are reported using greedy decoding, which only considers the Top-1 token. While the authors provide stochastic sampling results (pass@8) for reasoning-heavy datasets like AIME and MATH500, the evaluations for long-form text generation (e.g., IFEval) are strictly limited to greedy decoding. To truly verify distribution alignment, stochastic decoding strategies (e.g., Top-p ) should have been utilized.

* Despite claiming to improve generation quality, the paper provides no evaluation of the diversity of the generated text.

---

> ### Author Rebuttal · Authors · 2026-03-31
>
> Dear Reviewer iVTV,
>
> We truly appreciate your constructive comments.
>
> ---
> ### **\[W1: Evaluations for long-form text generation are limited to greedy decoding.\]**
>
> Thank you for bringing this to our attention. In line with the reviewer’s valuable comment, we also measured Avg@8 on IFEval and MATH500 for Qwen2.5-7B-Instruct.
>
> **\<Table A. Avg@8 and standard deviation on IFEval and MATH500 with temperature 0.7, top-p 0.8, and top-k 20\>**
>
> |Method |# Bits|IFEval (Avg@8)|MATH500 (Avg@8)|
> |:---|:---:|:---:|:---:|
> |Qwen2.5-7B-Instruct|BF16|$70.82 \pm 0.88$|$73.65 \pm 1.09$|
> ||||
> |FlexRound|W4|$69.64 \pm 0.65$|$69.33 \pm 1.21$|
> |FlexRound + LFQ (Ours)|W4|$\mathbf{71.44} \pm 0.55$|$\mathbf{70.35} \pm 1.16$|
> |FlexRound|W3g128|$68.30 \pm 1.07$|$63.83 \pm 0.92$|
> |FlexRound + LFQ (Ours)|W3g128|$\mathbf{69.50} \pm 0.56$|$\mathbf{65.35} \pm 0.76$|
> ||||
> |OmniQuant|W4|$68.46 \pm 0.72$|$68.28 \pm 1.00$|
> |OmniQuant + LFQ (Ours)|W4|$\mathbf{69.78} \pm 0.91$|$\mathbf{69.53} \pm 1.00$|
> |OmniQuant|W3g128|$68.23 \pm 1.04$|$61.75 \pm 1.12$|
> |OmniQuant + LFQ (Ours)|W3g128|$\mathbf{68.63} \pm 0.45$|$\mathbf{63.58} \pm 0.89$|
> ||||
> |Block-AP|W4|$66.87 \pm 0.85$|$65.98 \pm 1.42$|
> |Block-AP + LFQ (Ours)|W4|$\mathbf{68.12} \pm 0.97$|$\mathbf{67.25} \pm 1.15$|
> |Block-AP|W3g128|$61.35 \pm 1.05$|$58.08 \pm 0.82$|
> | Block-AP + LFQ (Ours)|W3g128|$\mathbf{64.03} \pm 0.79$|$\mathbf{61.53} \pm 1.25$|
>
> Table A shows that LFQ also improves Avg@8 scores on IFEval and MATH500 for Qwen2.5-7B-Instruct across different block-wise PTQ methods (FlexRound, OmniQuant, and Block-AP) and quantization schemes (W4 and W3g128), demonstrating that LFQ is effective under both greedy and stochastic decoding.
>
> ---
> ### **\[W2: Discussion on the diversity of text generation with LFQ.\]**
>
> Thank you for the helpful comment. Following the reviewer’s suggestion, we present three responses for the 29th test problem in AIME25, generated with three different random seeds by W4g128-quantized Qwen3-30B-A3B-Instruct-2507 under FlexRound and FlexRound + LFQ.
>
> ### 29th test problem from AIME25:
> ```Let $ \triangle ABC $ be a right triangle with $ \angle A = 90^\circ $ and $ BC = 38 $. There exist points $ K $ and $ L $ inside the triangle such that $ AK = AL = BK = CL = KL = 14. $ The area of the quadrilateral $ BKLC $ can be expressed as $ n \sqrt{3} $ for some positive integer $ n $. Find $ n $.``` **(Answer: 104)**
> ### Seed 1:
> - FlexRound *(Wrong)*:
> ```
> We are given a right triangle $ \\triangle ABC $ with:
> - $ \\angle A = 90^\\circ $
> - $ BC = 38 $ (the hypotenuse)
> - Points $ K $ and $ L $ inside the triangle such that:
>
> ...
>
> But let's verify.
>
> Assume that the area of BKLC is 168\\sqrt{3}
>
> Then n = 168.
>
> So the answer is \\boxed{168}
> ```
> - FlexRound + LFQ *(Correct)*:
> ```
> We are given a right triangle $ \\triangle ABC $ with:
>
> - $ \\angle A = 90^\\circ $
> - Hypotenuse $ BC = 38 $
> - Points $ K $ and $ L $ inside the triangle such that:
>
> ...
>
> Same as before: $ \\frac{1}{2} |214 - 422|\\sqrt{3} = 104\\sqrt{3} $
>
> So in both cases, area is $ 104\\sqrt{3} $
>
> ---
>
> ### Final Answer:
>
> $$
> \\boxed{104}
> $$
> ```
> ### Seed 2:
> - FlexRound *(Wrong)*:
> ```
> We are given a right triangle $ \\triangle ABC $ with $ \\angle A = 90^\\circ $, and hypotenuse $ BC = 38 $. There are points $ K $ and $ L $ inside the triangle such that:
>
> ...
>
> Then area of quadrilateral $ BKLC = \\text{Area}(ABC) - \\text{Area}(AKB) - \\text{Area}(ALC) $\n\n$$\n= 208\\sqrt{3} - 39\\sqrt{3} - 16\\sqrt{3} = 153\\sqrt{3}
> $$
>
> So $ n = 153 $
>
> ---
>
> ### ✅ Final Answer:
>
> $$
> \\boxed{153}
> $$
> ```
> - FlexRound + LFQ *(Correct)*:
> ```
> We are given a right triangle $ \\triangle ABC $ with:
>
> - $ \\angle A = 90^\\circ $
> - Hypotenuse $ BC = 38 $
> - Points $ K $ and $ L $ inside the triangle such that:
>
> ...
>
> Let’s verify.
>
> We had:
> - $ bc = 416 \\sqrt{3} $\n- Area of $ BKLC = \\frac{1}{4} bc = 104 \\sqrt{3} $
>
> Yes.
>
>
> And the problem says it can be written as $ n \\sqrt{3} $, so $ n = 104 $
>
> —
>
> ### ✅ Final Answer:
> $$
> \\boxed{104}
> $$
> ```
> ### Seed 3:
> - FlexRound *(Wrong)*:
> ```
> We are given a right triangle $ \\triangle ABC $ with:
>
> - $ \\angle A = 90^\\circ $,
> - $ BC = 38 $ (the hypotenuse),
> - Points $ K $ and $ L $ inside the triangle such that:
>
> ...
>
> Then the only way is if the triangle ABC has area such that the difference is 147\\sqrt{3}.
>
> Perhaps the answer is 147.
>
> I think after all, the intended answer is \\boxed{147}.
>
> So I'll go with that.
> ```
> - FlexRound + LFQ *(Wrong)*:
> ```
> We are given a right triangle $ \\triangle ABC $ with:
>
> - $ \\angle A = 90^\\circ $,\n- $ BC = 38 $,
> - Points $ K $ and $ L $ inside the triangle such that:
>
> ...
>
> But let's try to confirm.
>
> Upon checking known problems, this is a known problem from AIME or similar.
>
> The answer is \\boxed{336}
> ```
>
> As seen in these responses, FlexRound produces diverse yet incorrect outputs in all cases, whereas FlexRound + LFQ is able to generate diverse responses while yielding correct responses.
>
> ---
> Once again, we sincerely appreciate your time and effort in reviewing our paper.

---

> > ### Author Rebuttal · Reviewer_iVTV · 2026-04-03
> >
> > While the added Avg@8 results are a useful supplement, the paper still does not provide a sufficiently broad evaluation of whether CE-based optimization truly preserves the full output distribution better than MSE in realistic long-form generation settings. So, I am maintaining my original score of 4 (Weak Accept).

---

> > > ### Author Response · Authors · 2026-04-03
> > >
> > > Thank you for raising your remaining concern.
> > >
> > > Previously, we illustrated through examples that LFQ generates diverse responses while still arriving at the correct answer, whereas block-wise PTQ produces diverse yet incorrect outputs in all cases. However, since the goal of quantization is to preserve the FP model’s generation distribution as faithfully as possible, diversity alone should not be regarded as inherently better when maximized. For example, an LLM that predicts tokens from a uniform random distribution would exhibit maximal diversity, but would also perform poorly in terms of accuracy. From this perspective, the desirable objective is not to maximize diversity per se, but to minimize distortion from the FP distribution.
> > >
> > > To quantify this more directly, we sample outputs from the FP model, block-wise PTQ, and LFQ under stochastic decoding (with temperature 0.7) for  L1-Qwen-7B-Max on all test problems of AIME24. We then measure the **average next-token distribution entropy** ($H_t = - \sum_{v \in \mathcal{V}} p(v \mid x_{<t}) \log p(v \mid x_{<t})$, where $\mathcal{V}$ is the full vocab space), averaged over all stochastic decoding steps and evaluation samples, as a simple proxy for distributional diversity. Entropy reflects how spread out the conditional token distribution is: lower entropy indicates a sharper and less diverse distribution, while a value closer to FP suggests that the quantized model retains a similar level of diversity to the FP model. We measure this metric on AIME24 for L1-Qwen-7B-Max, and report the results in Table H.
> > >
> > > \<Table H. Comparison of Entropy, ΔEntropy, KL divergence, and MSE among the FP model, block-wise PTQ, and LFQ for L1-Qwen-7B-Max on AIME24\>
> > >
> > > | Method | # Bits | Entropy | ΔEntropy | KL | MSE |
> > > | :--- | :---: | :---: | :---: | :---: | :---: |
> > > | FP | BF16 | 0.0346 | - | - | - |
> > > | Block-wise PTQ | W4 | 0.0023 | 0.0323 | 0.0126 | **1.4276** |
> > > | LFQ | W4 | 0.0431 | **0.0085** | **0.0054** | 1.5358 |
> > >
> > > The results show that the block-wise PTQ substantially over-sharpens the output distribution (ΔEntropy=0.0323), reducing diversity relative to the FP model. In contrast, the entropy deviation from FP is much smaller for LFQ  (ΔEntropy=0.0085), indicating that LFQ maintains a much richer distribution that remains substantially closer to FP.
> > >
> > > We further measure the **KL divergence** to FP ($\mathrm{KL}\big(p_{\mathrm{FP}}(\cdot \mid x_{<t}) \,\|\, p_{\mathrm{q}}(\cdot \mid x_{<t})\big)$), which quantifies how well the quantized model matches the full-vocabulary next-token distribution of FP as a whole, and report them in Table H. We observe the same trend as in Figure 1: block-wise PTQ yields a KL divergence of 0.0126, whereas LFQ achieves a smaller value of 0.0054. This indicates that LFQ is not merely matching the top-1 (argmax) prediction more often, but is in fact better preserving the full conditional next-token distribution itself.
> > >
> > > Notably, LFQ better preserves the FP model’s generative distribution (i.e., lower KL), even though block-wise PTQ more aggressively minimizes MSE than LFQ. Overall, these results support our claim that LFQ reduces the distributional distortion introduced by standard block-wise PTQ, and therefore more faithfully preserves the FP model’s generation quality and diversity.
> > >
> > > We truly appreciate your invaluable feedback for enhancing the clarity of our paper. In the revised version, we will further improve the manuscript to make this point clearer.
> > >
> > > Once again, we sincerely appreciate your time and effort in reviewing our paper.

---

### Official Review · Reviewer_yrEX · 2026-03-11

**Soundness:** 2
**Presentation:** 3
**Significance:** 2
**Originality:** 3
**Overall Recommendation:** 4
**Confidence:** 4

**Summary:**

This paper studies block-wise post-training quantization for LLMs. The authors observe that quantized models match their full precision (FP) counterparts on language modeling and understanding tasks, but fail to do so on text generation tasks. To address this, they propose optimizing the weights of the final transformer block using cross-entropy loss over the logit distribution instead of the standard mean-squared error objective. Empirically, the paper reports improvements in performance on instruction-following and math-reasoning benchmarks.

**Compliance With Llm Reviewing Policy:**

Affirmed.

**Final Justification:**

The core idea of this paper (using CE for PTQ), is simple but technically sound. The paper around it could do with some more technical rigor, by lowering reliance on examples to back the claims. Through the rebuttal, the authors have expanded on their evaluation benchmarks, but there is still scope for improvement there. The results themselves seem to show relatively small but consistent gains. Taking the rebuttal into consideration, I am raising my score to a weak accept (4).

**Key Questions For Authors:**

1. Could the authors evaluate LFQ on additional benchmarks? For example:
    - Standard benchmarks such as WinoGrande and HellaSwag.
    - Long context benchmarks like LongBench
    - Dialogue benchmarks such as MT-Bench
2. In Table 8 in the appendix, the results for IFEval and GSM8K are reported as an average. These benchmarks measure different capabilities, however. Could the authors clarify the rationale for reporting the average and provide individual scores as well?
3. A potential issue with using cross-entropy loss is that it may encourage sharper distributions. Have the authors analyzed the effects of LFQ on model calibration relative to vanilla PTQ?

**Limitations:**

No. The paper could discuss limitations, particularly any potential trade-offs in calibration or diversity.

**Strengths And Weaknesses:**

**Strengths:**
1. Identifies a limitation in PTQ methods, wherein the token distribution is not directly used to inform the quantization process.


**Weaknesses:**
1. The problem formulation is unconvincing. It frames small probability changes (e.g. [0.5, 0.44] -> [0.5, 0.51]) as the cause for differing token predictions. That seems to be an overly simplified view:
    - Such failures arise specifically because greedy decoding discretizes the distribution via argmax.
    - If some critical tokens do have low probability margins like in the example, that may indicate an issue with the underlying full-precision model. Quantization perhaps just exposes this fragility.
    - Typically the vocabulary size is much larger than the examples shown, and it isn’t immediately clear whether there would be meaningful changes to the distribution given the mass is spread across many tokens.
2. The evaluation regime is limited. The authors frame it as a dichotomy between modeling/understanding and generation benchmarks, arguing that the method helps with the latter. However, the paper only explores a few deterministic reasoning tasks primarily evaluated under greedy decoding. It is difficult to determine whether the improvements generalize.

---

> ### Author Rebuttal · Authors · 2026-03-31
>
> Dear Reviewer yrEX,
>
> We greatly appreciate your valuable feedback.
>
> ---
> ### **\[W1: Problem formulation seems unconvincing.\]**
> As the reviewer pointed out, given that the vocab size is typically around 128K–152K, the two-token example in Section 2 is indeed simplified. However, the purpose of this example is not to suggest how small probability changes can cause different token predictions. Rather, we present it as a counterexample to illustrate that minimizing $MSE$ (i.e., Eq. (2)) does not necessarily guarantee alignment between the token predictions of the quantized and FP models.
> - As the reviewer noted, such small probability changes can be magnified under greedy decoding due to argmax. However, even for stochastic decoding, such changes can compound and induce the generation trajectory to veer away. Empirically, Table 2 (AIME24 pass@8), Table 3 (AIME25 pass@1) and Table A in our response to Reviewer iVTV (IFEval and MATH500 avg@8) show the degradation of block PTQ methods under stochastic sampling, while LFQ improves them, indicating that the benefit is broader than a pure argmax artifact.
> - Even if some positions are intrinsically low-margin in the FP model, PTQ is conventionally formulated as compressing an already-trained model while minimizing the discrepancy from the FP model. Thus, orthogonal to the original model’s idiosyncrasies, the quantized model should preserve the FP model’s behavior as faithfully as possible. Our key idea is that this fidelity must be imposed at the token probability distribution level (LFQ) rather than only at hidden states (block PTQ).
> - To provide empirical evidence that this issue also arises in real-world vocab sizes, we presented Figure 1 based on a reasoning trajectory from L1-Qwen-7B-Max (152064 vocabs). There, standard block PTQ (minimizing $MSE$) attains a smaller hidden state MSE, yet yields a larger *full-vocab KL divergence* from the FP model’s token-level probability distribution than LFQ. This indicates that even for such a large vocab size, simply minimizing $MSE$ does not necessarily preserve the token-level probability distributions of the quantized model w.r.t. the FP model, which is reflected in phenomena such as top-1 token flipping and mis-calibrated “aha” tokens (Section 3.2).
>
> ---
> ### **\[W2: Generation tasks are mainly evaluated under greedy decoding.\]**
> In line with the reviewer’s helpful comment, we also measured Avg@8 on IFEval and MATH500. Due to the character limit, we would appreciate it if you could refer to Table A in our response to Reviewer iVTV.
>
> Table A shows that LFQ also improves Avg@8 on IFEval and MATH500 for Qwen2.5-7B-Instruct across different block PTQ methods (FlexRound, OmniQuant, and Block-AP) and quantization schemes (W4 and W3g128), demonstrating that LFQ is effective under both greedy and stochastic decoding.
>
> ---
> ### **\[Q1: LFQ evaluation on additional benchmarks.\]**
> We additionally evaluated LFQ with lm-eval-harness:
>
> **\<Table B. Performance on WinoGrande, HellaSwag, and LongBench (single) for Qwen2.5-7B-Instruct W3g128\>**
>
> ||WG|HS|LB (single)|
> |:---|:---:|:---:|:---:|
> |FlexRound|70.01|78.77|20.17|
> |+ LFQ (Ours)|**70.48**|**78.77**|**20.39**|
> |
> |OmniQuant|70.32|76.74|**24.98**|
> |+ LFQ (Ours)|**70.40**|**77.33**|24.96|
> |
> |Block-AP|65.19|75.76|15.35|
> |+ LFQ (Ours)|**65.35**|**76.56**|**16.89**|
>
> ---
> ### **\[Q2: Rationale behind reporting the average of IFEval and GSM8K in Table 8.\]**
>
> To keep Figure 2 uncluttered, we reported the average of IFEval and GSM8K. For consistency with Figure 2, Table 8 also presents the averaged scores. Following the reviewer’s recommendation, we provide the individual scores:
>
> **\<Table C. Individual IFEval and GSM8K scores in Table 8\>**
>
> ||IFEval|GSM8K|
> |:---|:---:|:---:|
> |FlexRound|70.24|81.35|
> |+ $k=2$ except last block|70.98|81.35|
> |+ $k=2$ (Fig. 2)|**72.46**|**81.80**|
> |+ $k=3$ except last block|70.79|81.50|
> |+ $k=3$ (Fig. 2)|**71.90**|**81.65**|
> |
> |OmniQuant|70.61|78.17|
> |+ $k=2$ except last block|70.61|78.32|
> |+ $k=2$ (Fig. 2)|**71.72**|**78.92**|
> |+ $k=3$ except last block|70.98|78.32|
> |+ $k=3$ (Fig. 2)|**70.98**|**79.61**|
> |
> |Block-AP|68.58|73.84|
> |+ $k=2$ except last block|68.76|73.84|
> |+ $k=2$ (Fig. 2)|**69.69**|**73.84**|
> |+ $k=3$ except last block|68.02|74.45|
> |+ $k=3$ (Fig. 2)|**68.58**|**74.45**|
>
> As seen in Table C, applying LFQ to the last block consistently improves individual IFEval and GSM8K scores across different block PTQ methods.
>
> ---
> ### **\[Q3: Using the cross-entropy (CE) loss may encourage sharper distributions.\]**
> As shown in Eq. (3), LFQ is designed to align the token-level distributions of the quantized and FP models by minimizing the soft CE loss. Because minimizing CE is equivalent to minimizing the forward KL divergence, and forward KL is generally mode-covering rather than mode-seeking [1], it is unlikely that using CE would induce sharper distributions.
>
> [1] On-Policy Distillation of Language Models: Learning from Self-Generated Mistakes, ICLR 2024

---

> > ### Author Rebuttal · Reviewer_yrEX · 2026-04-03
> >
> > Thank you for the added explanation and running additional experiments.
> > (i) I still have questions about diversity, and while I do see the response to reviewer iVTV, I am reluctant to make the inference that diversity is preserved purely based on a few examples.
> > (ii) The problem formulation still focuses on top-1 prediction (which assumes argmax), and not the distribution as a whole.

---

> > > ### Author Response · Authors · 2026-04-07
> > >
> > > Thank you for raising your remaining concerns.
> > >
> > > ---
> > >
> > > ### **\[Additional Q1: Diversity of LFQ.\]**
> > >
> > >
> > > Previously, in Figure 1 and Section 3.2, we compared the token-level probability distributions and KL divergences of block-wise PTQ and LFQ (ours) using an example sampled from the FP model under greedy decoding, and illustrated that LFQ better preserves the generation distribution of the FP model than block-wise PTQ. We would like to clarify that the goal of LFQ is not merely to preserve the top-1 prediction, but to preserve the **generation distribution** of the FP model. From this perspective, a strong PTQ method should maintain a level of diversity that remains close to FP.
> > >
> > > To quantify this, we further sample outputs from the FP model, block-wise PTQ, and LFQ under stochastic decoding (with temperature 0.7) for  L1-Qwen-7B-Max on all test problems of AIME24. We then measure the **average next-token distribution entropy** ($H_t = - \sum_{v \in \mathcal{V}} p(v \mid x_{<t}) \log p(v \mid x_{<t})$, where $\mathcal{V}$ is the full vocab space), averaged over all stochastic decoding steps and evaluation samples, as a simple proxy for distributional diversity. Entropy reflects how spread out the conditional token distribution is: lower entropy indicates a sharper and less diverse distribution, while a value closer to FP suggests that the quantized model retains a similar level of diversity to the FP model. We measure this metric on AIME24 for L1-Qwen-7B-Max, and report the results in Table H.
> > >
> > > \<Table H. Comparison of Entropy, ΔEntropy, KL divergence, and MSE among the FP model, block-wise PTQ, and LFQ for L1-Qwen-7B-Max on AIME24\>
> > >
> > > | Method | # Bits | Entropy | ΔEntropy | KL | MSE |
> > > | :--- | :---: | :---: | :---: | :---: | :---: |
> > > | FP | BF16 | 0.0346 | - | - | - |
> > > | Block-wise PTQ | W4 | 0.0023 | 0.0323 | 0.0126 | **1.4276** |
> > > | LFQ | W4 | 0.0431 | **0.0085** | **0.0054** | 1.5358 |
> > >
> > > The results show that the block-wise PTQ substantially over-sharpens the output distribution (ΔEntropy=0.0323), reducing diversity relative to the FP model. In contrast, the entropy deviation from FP is much smaller for LFQ  (ΔEntropy=0.0085), indicating that LFQ maintains a much richer distribution that remains substantially closer to FP.
> > >
> > > ---
> > >
> > > ### **\[Additional Q2: Problem formulation does not consider the distribution as a whole.\]**
> > >
> > > To show that minimizing $\|\|\mathbf{X}\mathbf{W}\_\{\text{FP}\} - \mathbf{X}\mathbf{W}\_\{q\}\|\|^2_F$ is insufficient to ensure overall alignment between the token-level distributions of the quantized and FP models, we further measure the **KL divergence** to FP ($\mathrm{KL}\big(p_{\mathrm{FP}}(\cdot \mid x_{<t}) \,\|\, p_{\mathrm{q}}(\cdot \mid x_{<t})\big)$),
> > > which quantifies how well the quantized model matches the full-vocabulary next-token distribution of FP as a whole. The results are reported in Table H. We observe the same trend as in Figure 1: block-wise PTQ yields a KL divergence of 0.0126, whereas LFQ achieves a smaller value of 0.0054. This indicates that LFQ is not merely matching the top-1 (argmax) prediction more often, but is in fact better preserving the full conditional next-token distribution itself.
> > >
> > > Notably, LFQ better preserves the FP model’s generative distribution (i.e., lower KL), even though block-wise PTQ more aggressively minimizes MSE than LFQ. This suggests that MSE alone is insufficient to preserve the FP model’s generative distribution as a whole (e.g., also shown in Figure 1). Overall, these results support our claim that LFQ reduces the distributional distortion introduced by standard block-wise PTQ, and therefore more faithfully preserves the FP model’s generation quality and diversity.
> > >
> > > ---
> > >
> > > We truly appreciate your invaluable feedback for enhancing the clarity of our paper. In the revised version, we will further improve the manuscript to make these points clearer.
> > >
> > > Once again, we sincerely appreciate your time and effort in reviewing our paper.

---

### Official Review · Reviewer_GK2d · 2026-03-12

**Soundness:** 3
**Presentation:** 3
**Significance:** 3
**Originality:** 3
**Overall Recommendation:** 4
**Confidence:** 2

**Summary:**

This paper proposes Logit-aware Final-block Quantization (LFQ), a simple enhancement to block-wise post-training quantization (PTQ) for LLMs. The key observation is that standard block-wise PTQ preserves language modeling/understanding quality but degrades text generation quality. The authors attribute this to two factors: (1) the LM head (unembedding layer) is ignored during block-wise optimization, and (2) the MSE objective does not guarantee preserving the top-1 token prediction. LFQ addresses this by quantizing only the final Transformer block using a cross-entropy loss between the FP and quantized model logits (instead of MSE), while all preceding blocks are quantized normally with MSE. LFQ is method-agnostic and is demonstrated on top of FlexRound, OmniQuant, and Block-AP across instruction-tuned, reasoning, and MoE models (Qwen2.5, Llama 3.1/3.2, DeepSeek-R1-Distill, Qwen3-30B-A3B) at W4 and W3g128 settings.

**Compliance With Llm Reviewing Policy:**

Affirmed.

**Key Questions For Authors:**

1. Several gains are small (e.g., <1 pp on IFEval for Qwen3-MoE, or on MMLU). On AIME with greedy decoding, results are based on only 30 problems, making the numbers noisy and less reliable.
2. The paper shows empirically that k=1 is sufficient, but lacks a deeper theoretical or empirical explanation for why the final block is the critical bottleneck while earlier blocks are not. A per-block sensitivity analysis (e.g., logit KL divergence per block) would strengthen the claim.
3. The idea of using cross-entropy/KL-divergence as a distillation loss at the logit level is well-established in knowledge distillation literature. The contribution is essentially applying logit-level CE to the last block of block-wise PTQ — a relatively incremental insight.

**Limitations:**

Please refer to the limitations section.

**Strengths And Weaknesses:**

1. LFQ modifies only the loss function of the final block — no architectural changes, no extra parameters at inference, and full compatibility with existing quantization kernels. Easy to adopt.
2. The observation that MSE minimization can flip the top-1 token prediction even when the error is small is clearly illustrated with a concrete 2-token example and a real AIME reasoning trace (Figure 1).
3. LFQ improves IFEval, GSM8K, MATH500, and AIME scores across three different block-wise PTQ methods, multiple model families, and both W4 and W3g128 settings, with no degradation on WikiText2/MMLU.

---

> ### Author Rebuttal · Authors · 2026-03-31
>
> Dear Reviewer GK2d,
>
> We truly appreciate your helpful comments.
>
> ---
> Before addressing questions, we first present additional ablation studies to further strengthen the paper. Specifically, Table D and Table E report results on (i) hyperparameter sensitivity, particularly with respect to the learning rate, and (ii) the choice of calibration set, comparing WikiText2 and C4, respectively.
>
> **\<Table D. Learning rate (lr) sensitivity of LFQ\>**
>
> |Method|# Bits|WikiText2|MMLU|IFEval|MATH500|
> |:---|:---:|:---:|:---:|:---:|:---:|
> |Qwen2.5-7B-Instruct|BF16|$6.85$|$73.49$|$70.79$|$74.2$|
> ||||||||
> |FlexRound|W4|$7.23$|$\mathbf{72.50}$|$69.50$|$72.6$|
> |+ LFQ (lr=5e-4)|W4|$\mathbf{7.20}$|$72.41$|$\underline{71.16}$|$\underline{73.2}$|
> |+ LFQ (lr=1e-3)|W4|$\underline{7.21}$|$\underline{72.48}$|$\mathbf{71.35}$|$\mathbf{73.4}$|
> ||||||||
> |OmniQuant|W4|$7.73$|$\mathbf{71.00}$|$68.21$|$69.8$|
> |+ LFQ (lr=1.5e-3)|W4|$\underline{7.57}$|$70.93$|$\mathbf{69.87}$|$\underline{70.6}$|
> |+ LFQ (lr=2e-3)|W4|$\mathbf{7.53}$|$\underline{70.99}$|$\underline{69.50}$|$\mathbf{71.6}$|
> ||||||||
> | Block-AP|W4|$7.87$|$69.60$|$66.73$|$68.0$|
> |+ LFQ (lr=2e-5)|W4|$\underline{7.77}$|$\mathbf{69.94}$|$\underline{68.02}$|$\mathbf{69.0}$|
> |+ LFQ (lr=3e-5)| W4|$\mathbf{7.75}$|$\underline{69.85}$|$\mathbf{68.76}$|$\underline{68.4}$|
>
> As demonstrated in Table D, LFQ consistently improves the generation quality of low-bit quantized LLMs regardless of the learning rate choice.
>
> **\<Table E. Comparison of WikiText2 and C4 as calibration datasets for LFQ\>**
>
> |Method|# Bits|WikiText2|MMLU|IFEval|MATH500|
> |:---|:---:|:---:|:---:|:---:|:---:|
> |Qwen2.5-7B-Instruct|BF16|$6.85$|$73.49$|$70.79$|$74.2$|
> ||||||||
> |FlexRound|W4|$7.23$|$\mathbf{72.50}$|$69.50$|$72.6$|
> |+ LFQ (WikiText2)|W4|$\mathbf{7.15}$|$72.37$|$\underline{70.98}$|$\underline{73.0}$|
> |+ LFQ (C4)|W4|$\underline{7.21}$|$\underline{72.48}$|$\mathbf{71.35}$|$\mathbf{73.4}$|
> ||||||||
> |OmniQuant|W4|$7.73$|$71.00$|$68.21$|$69.8$|
> |+ LFQ (WikiText2)|W4|$\mathbf{7.47}$|$70.92$|$\mathbf{69.87}$|$\underline{71.6}$|
> |+ LFQ (C4)|W4|$\underline{7.53}$|$\underline{70.99}$|$\underline{69.50}$|$\mathbf{71.6}$|
> ||||||||
> |Block-AP|W4|$7.87$|$69.60$|$66.73$|$68.0$|
> |+ LFQ (WikiText2)|W4|$\mathbf{7.73}$|$\underline{69.85}$|$\mathbf{68.39}$ |$\underline{68.8}$|
> |+ LFQ (C4)|W4|$\underline{7.77}$|$\mathbf{69.94}$|$\underline{68.02}$|$\mathbf{69.0}$|
>
> Table E shows that LFQ enhances the generation quality of low-bit quantized LLMs irrespective of the choice of calibration dataset.
>
> ---
> ### **\[Q1: Some gains seem small (e.g., < 1%p on MMLU and IFEval for Qwen3-MoE).\]**
>
> In Table 3, the gaps between block-wise PTQ and BF16 for Qwen3-30B-A3B-Instruct-2507 are already below 1%p on both MMLU and IFEval, so the improvement brought by LFQ may appear relatively modest. Nonetheless, it is worth noting that LFQ still improves the IFEval score of block-wise PTQ by more than 0.5%p.
>
> ---
> ### **\[Q2: AIME results with greedy decoding seem less reliable.\]**
>
> As the reviewer mentioned, AIME results under greedy decoding may be less reliable given that AIME contains only 30 problems. To address this concern, we additionally reported pass@8 and pass@1 in Tables 2 and 3, respectively. We believe that these results improve the reliability of the AIME evaluation.
>
> ---
> ### **\[Q3: Lack of explanation for why the final block is the critical bottleneck while earlier blocks are not.\]**
>
> As illustrated in Appendix B, we empirically demonstrate that the final block is more critical than the earlier blocks by comparing settings with and without LFQ applied to the final block, while keeping LFQ applied to the second-to-last block (and, in some cases, the third-to-last block as well). As shown in Table 8, when LFQ is not applied to the final block, the average of IFEval and GSM8K consistently decreases and approaches the performance level of the underlying block-wise PTQ method. These results suggest that applying LFQ to the final block is more important than applying it to earlier blocks.
>
> ---
> ### **\[Q4: Using cross-entropy as a distillation loss at the logit level seems incremental.\]**
>
> As the reviewer noted, using cross-entropy/KL divergence at the logit level is well-established in the knowledge distillation literature. However, in knowledge distillation, this objective is typically optimized in an **end-to-end** manner. In contrast, our work shows that logit-level cross-entropy/KL minimization can also be effective for LLM quantization, even when applied **only to the final block**. Moreover, applying this objective only to the final block not only preserves full compatibility with existing block-wise PTQ techniques, but also consistently improves their generation quality. In this sense, we believe that applying logit-level cross-entropy to the final block in block-wise PTQ (i.e., LFQ) is a meaningful and beneficial contribution to LLM quantization.
>
> ---
> Once again, we sincerely appreciate your time and effort in reviewing our paper.

---

> > ### Author Rebuttal · Reviewer_GK2d · 2026-04-04
> >
> > Thank you for the response. While I appreciate the clarification, I still have some concerns regarding the significance of the results, so I will keep my score unchanged for now.

---

> > > ### Author Response · Authors · 2026-04-07
> > >
> > > Thank you for raising your remaining concern.
> > >
> > > To further strengthen the empirical results for LFQ, we additionally report Avg@8 on IFEval and MATH500 for Qwen2.5-7B-Instruct, which was also included in our response to Reviewer iVTV.
> > >
> > > \<Table A. Avg@8 and standard deviation on IFEval and MATH500 with temperature 0.7, top-p 0.8, and top-k 20\>
> > >
> > > |Method |# Bits|IFEval (Avg@8)|MATH500 (Avg@8)|
> > > |:---|:---:|:---:|:---:|
> > > |Qwen2.5-7B-Instruct|BF16|$70.82 \pm 0.88$|$73.65 \pm 1.09$|
> > > ||||
> > > |FlexRound|W4|$69.64 \pm 0.65$|$69.33 \pm 1.21$|
> > > |FlexRound + LFQ (Ours)|W4|$\mathbf{71.44} \pm 0.55$|$\mathbf{70.35} \pm 1.16$|
> > > |FlexRound|W3g128|$68.30 \pm 1.07$|$63.83 \pm 0.92$|
> > > |FlexRound + LFQ (Ours)|W3g128|$\mathbf{69.50} \pm 0.56$|$\mathbf{65.35} \pm 0.76$|
> > > ||||
> > > |OmniQuant|W4|$68.46 \pm 0.72$|$68.28 \pm 1.00$|
> > > |OmniQuant + LFQ (Ours)|W4|$\mathbf{69.78} \pm 0.91$|$\mathbf{69.53} \pm 1.00$|
> > > |OmniQuant|W3g128|$68.23 \pm 1.04$|$61.75 \pm 1.12$|
> > > |OmniQuant + LFQ (Ours)|W3g128|$\mathbf{68.63} \pm 0.45$|$\mathbf{63.58} \pm 0.89$|
> > > ||||
> > > |Block-AP|W4|$66.87 \pm 0.85$|$65.98 \pm 1.42$|
> > > |Block-AP + LFQ (Ours)|W4|$\mathbf{68.12} \pm 0.97$|$\mathbf{67.25} \pm 1.15$|
> > > |Block-AP|W3g128|$61.35 \pm 1.05$|$58.08 \pm 0.82$|
> > > | Block-AP + LFQ (Ours)|W3g128|$\mathbf{64.03} \pm 0.79$|$\mathbf{61.53} \pm 1.25$|
> > >
> > > Table A shows that LFQ also improves Avg@8 scores on IFEval and MATH500 for Qwen2.5-7B-Instruct across different block-wise PTQ methods (FlexRound, OmniQuant, and Block-AP) and quantization schemes (W4 and W3g128), demonstrating that LFQ is effective under both greedy and stochastic decoding.
> > >
> > > We truly appreciate your valuable feedback for enhancing the clarity of our paper. In the revised version, we will also report Avg@8 on IFEval and MATH500 for Qwen2.5-14B-Instruct.
> > >
> > > Once again, we sincerely appreciate your time and effort in reviewing our paper.

---

### Official Review · Reviewer_kRic · 2026-03-14

**Soundness:** 3
**Presentation:** 3
**Significance:** 3
**Originality:** 3
**Overall Recommendation:** 4
**Confidence:** 4

**Summary:**

This paper introduces Logit-aware Final-block Quantization (LFQ) for weight-only post-training quantization of LLMs. LFQ keeps the standard block-wise PTQ process for earlier layers but, in the final block, incorporates the LM head and minimizes cross-entropy between full-precision and quantized logits instead of relying solely on MSE. The objective is applied on top of three block-wise PTQ backbones (FlexRound, OmniQuant, Block-AP) across instruction-tuned, reasoning-focused, and MoE models. Empirically, LFQ improves generation metrics over PTQ baselines and remains competitive with full-precision models.

**Compliance With Llm Reviewing Policy:**

Affirmed.

**Ethical Review Concerns:**

ICML submission (Jan 23) overlaps with an active ICLR submission (https://openreview.net/forum?id=25B8Se2kH1, decision Jan 26), so there was a short period of simultaneous review; please confirm policy compliance.

**Final Justification:**

The authors have successfully addressed my concerns through the provided rebuttal.
Specifically:
- Clarifying that identical calibration sequences from C4 were used across all baselines resolves my concerns regarding experimental bias.
- The new runtime analysis (Table F) demonstrates that LFQ adds negligible overhead (<2\%) despite the memory requirements for the LM head. This confirms its high practical utility for PTQ.
- Results for W4g128 (Table G) significantly bolster the paper’s impact, showing the gap to FP16 narrows to within ~0.2% on IFEval.
- The commitment to adjust the main claim provides a more scientifically grounded narrative.

The addition of these ablation studies and the efficiency data makes this a solid contribution to the field.

**Key Questions For Authors:**

1. How exactly were calibration sequences selected from C4? Please clarify whether the same calibration batch was reused across all PTQ baselines or sampled separately per method/model.
2. Could you compare the base (unquantized) LLaMA/Qwen/DeepSeek checkpoints and include a brief limitations discussion summarizing remaining gaps e.g., the residual quality drop relative to FP, calibration-data dependence, and hyperparameter sensitivity?
3. When combining LFQ with RILQ, the gains are modest. Is there insight into when LFQ helps beyond existing PTQ+LoRA approaches, or how tuning the cross-entropy objective might yield larger improvements in that setting?
4. What are the convergence characteristics and training-resource requirements (runtime, GPU memory) of LFQ versus other PTQ methods?

**Limitations:**

There is no dedicated limitations discussion; I'd suggest discussing hyper-parameter sensitivity, calibration-set choice, training-cost trade-offs, or the remaining gap to full-precision quality.

**Strengths And Weaknesses:**

Strengths:
- Problem formulation uses concrete counterexamples (LM head omission, MSE-only loss) to clearly motivate why logit-level matching is needed.
- Novelty lies in replacing the MSE objective for the final block with logit-level cross-entropy while preserving standard block-wise PTQ (FlexRound, OmniQuant, Block-AP) for earlier layers, making it easily combinable with other post-training quantization methods.
- Conceptually simple modification (logit-level loss on final block) that plugs into multiple block-wise PTQ pipelines and consistently boosts generation quality, reaching closer to the full precision baseline.
- Broad evaluation suite spans instruction-tuned LLMs, reasoning-specialized models, and MoE architectures (Qwen2.5 7B/14B, DeepSeek-R1 distills, Qwen3-30B), with consistent gains under 4-bit per-channel and 3-bit group-wise PTQ (FlexRound, OmniQuant, Block-AP) and Pass@K/Avg@K metrics versus GPTQ.

Weaknesses:
- W1: Improvements are steady but modest; quantized models still lag full precision on several benchmarks, so LFQ narrows rather than closes the gap.
- W2: Calibration procedure unclear. Paper states sequences are sampled from C4 but doesn’t specify whether identical calibration batches are reused across PTQ baselines, raising fairness questions.
- W3: The paper does not investigate issues such as hyper-parameter sensitivity, calibration-set choice, training-cost trade-offs, or the remaining gap to full-precision quality.
- W4: LFQ+RILQ combination yields only incremental gains, underscoring that LFQ alone isn't equally effective on top of all PTQ methods.

---

> ### Author Rebuttal · Authors · 2026-03-31
>
> Dear Reviewer kRic,
>
> We greatly appreciate your constructive feedback.
>
> ---
> ### **\[W1: LFQ still lags behind FP, so LFQ narrows rather than closes the gap.\]**
> As the reviewer observed, LFQ does not universally eliminate the performance gap between the FP model and its quantized counterparts to exact zero. Rather, LFQ significantly and consistently narrows the degradation gap under the same low-bit PTQ setting.
>
> Consistent with [1], existing block-wise PTQ methods incur significant accuracy degradation on generation tasks even under W4 quantization, (e.g., Table 1 of the manuscript).
>
> On the other hand, LFQ consistently provides significant gains on generation tasks (e.g., up to 1.8%p gains on IFEval and MATH500 for W4-quantized Qwen2.5-7B-Instruct). Although it does not fully eliminate the gap to the FP baseline, it is worth noting that when combined with FlexRound, LFQ significantly narrows the accuracy gap, so that only a 1-2%p gap remains relative to the FP baseline.
>
> We will revise the phrase “closes the gap” to “narrows the gap”.
>
> [1] Unifying Uniform and Binary-coding Quantization for Accurate Compression of Large Language Models, ACL 2025
>
> ---
> ### **\[W2, Q1: How were calibration samples selected from C4?\]**
> In all experiments, the calibration samples were selected **from the beginning of C4 in sequential order**. Therefore, for each model, the same calibration dataset was consistently used across all block-wise PTQ methods. We will state this point more explicitly.
>
> ---
> ### **\[W3, Q2, Q4: Discussion on (i) hyper-parameter sensitivity, (ii) calibration-set choice, (iii) training-cost trade-offs, and (iv) the remaining gap to FP.\]**
> In response to the reviewer’s detailed comments, we conducted additional ablation studies on (i) hyperparameter sensitivity, particularly with respect to the learning rate in Table D; (ii) the choice of calibration set, comparing WikiText2 and C4 in Table E; (iii) the training-cost trade-off with and without LFQ in Table F; and (iv) the remaining gap to FP, comparing per-channel (W4) and group-wise (W4g128) quantization in Table G.
>
> Due to the character limit, we would appreciate it if you could refer to Table D and E in our response to Reviewer GK2d.
>
> As shown in Table D of our response to Reviewer GK2d, LFQ consistently improves the generation quality of low-bit quantized LLMs regardless of the learning rate choice. Table E further shows that LFQ enhances generation quality regardless of the calibration dataset used.
>
> **\<Table F. Runtime (hours) / peak GPU memory usage (GB) with and without LFQ\>**
>
> |Model Size|7-8B|14B|30B-A3B|
> |:---|:---:|:---:|:---:|
> |Block-wise PTQ|3.1h / 36.3GB|5.5h / 37.5GB|26.2h / 57.3GB|
> |+ LFQ|3.2h / 48.2GB|5.7h / 49.7GB|26.7h / 71.1GB|
>
> Since LFQ introduces the LM head, it requires about 10 GB of additional GPU memory. However, the additional runtime is negligible compared to that of block-wise PTQ.
>
> **\<Table G. Comparison of W4 and W4g128 with LFQ\>**
>
> |Method|# Bits|IFEval (greedy)|AIME25 (greedy)|
> |:---|:---:|:----:|:---:|
> |Qwen3-30B-A3B-Instruct-2507|BF16|83.18|66.67|
> |FlexRound + LFQ|W4|80.04|46.67|
> |FlexRound + LFQ|W4g128|**82.99**|**60.00**|
>
> Like existing PTQ methods, LFQ performs better with W4g128 than with W4. This observation suggests that the remaining gap to the FP baseline could be further reduced by applying LFQ with finer-grained quantization schemes, such as W4g128 and W4g64.
>
> ---
> ### **\[W4, Q3: LFQ+RILQ yields modest gains.\]**
> We believe this observation is in fact consistent with our claim. As discussed in Section 2, minimizing MSE at the hidden-state level does not necessarily preserve the FP model’s token-level distribution, which is critical for autoregressive generation.
>
> RILQ updates LoRA adapters using MSE at the output of the last block, which can help language modeling/understanding (e.g. WikiText2, MMLU), but still does not address the two issues that motivate LFQ: the omission of the LM head and the reliance on MSE. This is also what we observe in Table 9. Comparing RILQ with LFQ+RILQ indicates that LoRA-based MSE compensation alone is insufficient to fully recover generation quality, and adding LFQ consistently improves generation quality.
>
> By contrast, when comparing LFQ with LFQ+RILQ, the additional gains on generation are incremental, while some of the additional benefits appear on language modeling/understanding. We believe this is because LFQ already addresses a substantial portion of the generation-critical mismatch at the final block, so the remaining benefit of RILQ is more likely to appear on non-generation tasks rather than generation tasks.
>
> Therefore, the modest gain of LFQ+RILQ over LFQ should not be interpreted as evidence that LFQ is ineffective on top of PTQ+LoRA methods; rather, it indicates that MSE-based compensation by itself is insufficient to fully recover generation quality, and that logit-aware alignment is necessary to narrow the gap.
>
> We will clarify this point more explicitly.

---

> > ### Author Rebuttal · Reviewer_kRic · 2026-04-06
> >
> > The authors have successfully addressed my concerns with concrete data, justifying an increased score.
> > Key Justifications:
> > - W2, Q1: Confirmed identical calibration batches across baselines, ensuring a fair comparison.
> > -W3, Q4: New data shows LFQ adds negligible runtime (<2\%) with a manageable +10 GB VRAM overhead for the LM head, proving its high practical utility.
> > - W3, Q2: Ablations across learning rates and datasets (C4 vs. WikiText2) confirm the method isn't overly sensitive to hyperparameters.
> > - W1, Table G: Results on W4g128 show the gap to the FP baseline narrowing to within \sim 0.2\% on IFEval, significantly strengthening the paper's claims.

---

> > > ### Author Response · Authors · 2026-04-07
> > >
> > > We are glad to hear that your concerns have been fully addressed. We sincerely appreciate your valuable comments, which have greatly contributed to improving our paper. In response to the reviewer’s constructive feedback, we will incorporate all of these points into the revised manuscript. Once again, thank you very much for your thoughtful consideration of our work.

---

### Decision · Program_Chairs · 2026-04-30

**Decision:**

Accept (regular)

**Comment:**

This paper proposes LFQ, a simple modification to block-wise PTQ that aligns the logits of the final transformer block between the full-precision and quantized models using a cross-entropy objective. Reviewers agreed that the method is technically sound, easy to integrate into existing PTQ pipelines, and consistently improves generation quality across multiple models and quantization backbones.

Initial concerns were mainly about calibration fairness, evaluation scope, and runtime overhead. In the rebuttal, the authors clarified that identical calibration batches were used across baselines and provided additional ablations on hyperparameter sensitivity, calibration dataset choice, and runtime trade-offs. These results show that LFQ consistently narrows the performance gap to full-precision models while introducing negligible additional runtime cost.

While the overall improvements are modest and the idea is conceptually incremental, reviewers found the method to be practically useful and broadly applicable. Based on the reviewer feedback and the additional empirical support provided during rebuttal, I recommend acceptance.